# Monitoring ocean currents during the passage of Typhoon Muifa using optical-fiber distributed acoustic sensing

Jianmin Lin [1,2] ✉, Sunke Fang [1], Runjing He[1], Qunshu Tang [1,2], Fengzhong Qu[1,2], Baoshan Wang [3,4] ✉ & Wen Xu [1,2] ✉

In situ observations under typhoon conditions are sparse and limited. Distributed acoustic sensing (DAS) is an emerging technology that uses submarine optical-fiber (OF) cables to monitor the sea state. Here, we present DAS-based ocean current observations when a super typhoon passed overhead. The microseismic noise induced by ocean surface gravity waves (OSGWs) during Typhoon Muifa (2022) is observed in the ~0.08–0.38 Hz frequency band, with high-frequency (>0.3 Hz) component being tidally modulated. The OSGW propagation along the entire cable is successfully revealed via frequency–wavenumber analysis. Further, a method based on the current-induced Doppler shifts of DAS-recorded OSGW dispersions is proposed to calculate both speeds and directions of horizontal ocean currents. The measured current is consistent with the tidally induced sea-level fluctuations and sea-surface winds observed at a nearby ocean buoy. These observations demonstrate the feasibility of monitoring the ocean current under typhoon conditions using DAS-instrumented cables.

Cyclone systems, especially the well-known tropical cyclones (TCs), are devastating events that often have tremendous economic and societal impacts. However, accurately determining TC development, particularly TC intensity, is still a challenge due to the lack of sufficient in situ observations[1]. While the Dvorak technique[2,3], which is empirically based on cloud patterns and the infrared cloud-top temperature, is widely considered the best available tool to determine TC intensity from satellite imagery, its accuracy is unavoidably contaminated by rainfalls, clouds, breaking waves, and spray[4]. In situ measurements, such as aircraft reconnaissance and ocean buoys, are effective in reducing TC intensity and position uncertainties[5]. For example, Wang et al. [6] recently used drifter current measurements to estimate the sea-surface wind speed, and identified a positive trend of 1.8 m s⁻¹ per decade globally in the weak TC intensity during the 1991–2020 period. However, such in situ hydrological observations

remain sparse and limited due to the extreme sea state under TC conditions.

Ocean microseismic noise (typically in ~0.05–0.5 Hz), which is the most energetic component of the seismic noise spectrum mainly originating from ocean surface gravity waves (OSGWs; and hence wind fields) on the sea surface[7,8], has been long identified as a potential proxy of the sea state and any associated disturbances, including the tracking and monitoring of TCs over the oceans, thereby providing complementary observations to traditional atmospheric, oceanographic, and satellite observations[9–13]. Gutenberg[14] suggested that microseisms could be used in weather forecasting, and Grevemeyer et al.[15] reconstructed the wave climate in the northeast Atlantic Ocean using a 40-year record of wintertime microseisms at Hamburg, Germany. TCs over the oceans have been remotely located and tracked via microseism-based techniques, such as array beamforming[16–20].

[1]Key Laboratory of Ocean Observation-Imaging Testbed of Zhejiang Province, Zhejiang University, Zhoushan 316021, China. [2]Donghai Laboratory, Zhoushan 316021, China. [3]Deep Space Exploration Laboratory/School of Earth and Space Sciences, University of Science and Technology of China, Hefei 230026, China. [4]Mengcheng National Geophysical Observatory, University of Science and Technology of China, Hefei 230026, China. ✉e-mail: jmlin@zju.edu.cn; bwgeo@ustc.edu.cn; wxu@zju.edu.cn

However, most of the relevant studies have been limited to far-field observations at terrestrial broadband seismic stations due to the sparsity of ocean-bottom instrumentation. Davy et al.[9] analyzed the microseismic noise from a network of 57 broadband ocean bottom seismometers (OBSs) to infer the evolution of TC Dumile (2013) over the southwestern Indian Ocean, and demonstrated that seafloor microseisms can be used for real-time tracking and monitoring of major storms. However, accurate tracking and detailed near-field investigations of the TC are mainly limited by the complex excitation process and propagation of TC-induced microseisms, as well as the finite number and density of OBSs (average distance of 200 km between OBSs).

Distributed acoustic sensing (DAS) is an emerging technology that can provide strain measurements and effectively turn optical-fiber (OF) cables into dense seismo-acoustic arrays[21], thereby motivating many recent advances in microseism research[22,23] and ocean observations[24-30]. Lindsey et al.[24] mapped a number of unknown seafloor faults in Monterey Bay via seismic-wave propagation from a minor earthquake that was recorded by a submarine DAS-instrumented OF cable. The recorded microseismic noise and hydrodynamic signals were associated with OSGWs, storm-induced sediment transport, infragravity waves, and breaking internal waves, thereby demonstrating the potential of this optical-fiber sensing method for marine geophysics. Taweesintananon et al.[29] observed several types of ocean-bottom signals in the infrasound band (0.01–20 Hz) using a DAS-instrumented cable, and theoretically interpreted their generation mechanisms, including OSGW-loading seafloor pressure fluctuations and seismo-acoustic signals. Williams et al.[26] presented the potential application of ocean-bottom DAS in physical oceanography by analyzing OSGW propagation at a qualitative level to roughly estimate the current speed along the cable based on the DAS frequency–wavenumber ($f$–$k$) spectrum. Subsequently, Williams et al.[28] recovered spatiotemporal variations in current speed along the cable via DAS-based surface gravity wave interferometry. More recently, Mata Flores et al.[31] measured the deep-sea current speed with DAS-recorded vortex-induced vibration of suspended cable segments. However, these DAS-based measurements are solely geared towards ocean current speeds and have not yet been able to estimate the current directions, and the potential of utilizing DAS measurements as dense arrays remains to be further evaluated.

Here we report the in situ observations of microseismic noise via a pre-existing submarine DAS-instrumented OF cable in a complex geo-environment of an archipelago during the passage of a super typhoon. These observations show that the ocean-bottom DAS-instrumented cable is sensitive to typhoon-induced microseismic noise, even at low frequencies (<0.1 Hz), and effective in deriving information on OSGW along the cable. We further presented a DAS-based measurement method, which is based on two specific cable segments with different orientations, to successfully reveal both speeds and directions of the horizontal ocean-current component at high spatiotemporal resolutions during such an extreme weather event.

## Results and discussion
### Experiment overview
Typhoon Muifa (2022; known as Typhoon Inday in the Philippines) passed directly over a distributed acoustic sensing observatory (DASO), which utilized a pre-existing ocean-bottom OF cable in the Zhoushan Archipelago, off the northeastern coast of Zhejiang Province, eastern China (Fig. 1). The cable was installed in 2014 and is operated by China Mobile Communications Group. It is about 20 km long, connecting Daishan and Zhoushan islands along a relatively straight route. The detailed survey of the exact cable layout was not authorized due to security considerations. As an alternative, the water

depth along the cable was derived from a single-channel seismic profile that was acquired using a 2000-J Squid 2000 Sparker source (Applied Acoustic Engineering; United Kingdom) onboard a boat traveling at about 5.0 knots.

The DAS system, which was developed by the University of Science and Technology of China, was configured to probe phase changes of Rayleigh backscattering with 500-Hz sampling rate and 2-m gauge length. The used cable was spatially sampled at a 2-m channel interval along the cable, creating 9780 simultaneously recording sensors. However, the first 1125 channels are subaerial on the Daishan island, leaving 8655 channels (corresponding to 17.3 km of the cable; solid blue line in Fig. 1b) distributed beneath the sea surface. The raw DAS-recorded phase change data (with a unit of rad s⁻¹) was further processed into 2 Hz strain rate data with a unit of ε s⁻¹ (See detailed processing steps in Methods). A nearby ocean buoy (with minimum distance of ~1587 m to the cable; Supplementary Fig. 1) provided a continuous and accurate record of the sea-surface winds. These in situ observations during the passage of an ocean storm provided the unique opportunity to investigate the potential of utilizing DAS along pre-existing seafloor cables to monitor the sea state during such extreme weather conditions.

### Microseismic noise from ocean surface gravity waves
The microseismic noise recorded by the DASO is first revealed in a distance–frequency spectrogram (Fig. 2a). The spectrogram consists of the upper-quartile spectral power at successive individual channels along the entire cable from 4:10 UTC on September 13 to 12:00 UTC on September 15. The dominant spectral energy is at ~0.2 Hz for most of the channels, with both the amplitude and bandwidth of the spectra being inversely proportional to the water depth along the cable. Notable high-frequency (>0.3 Hz) microseismic noise (HFMN) is distinct where the water depth is shallower than ~15 m. This is consistent with the linear theory of OSGWs, whereby the seafloor pressure perturbation $\Delta P$ induced by an OSGW of height $\zeta$ decreases exponentially with water depth $h$ (Supplementary Fig. 2), i.e., $\Delta P(h, \zeta) = \frac{\rho g \zeta}{\cosh(2\pi k h)}$, where $\rho$ is the water density and $g$ is gravitational acceleration[25]. It is worth mentioning that the spectral amplitudes along the first ~3 km of the cable are exceptionally weak as a result of poor coupling between the cable and seafloor due to the rolling topography.

Mean time–frequency spectrograms are calculated at each set of four successive channels along the entire cable to further investigate the temporal evolution of the microseismic noise. The most prominent feature in the relatively shallow-water zone (e.g., channels 1253–1256 and 4068–4071) is that the HFMN is only observed during low tide (Fig. 2b, d). A similar phenomenon has been observed by Williams et al.[28] and Taweesintananon et al.[29] This can basically be interpreted as tidal modulation of the OSGW seafloor pressure. Crawford et al.[32] indicated that the theoretical maximum frequency of the OSGW seafloor pressure, $f_{max}$, can be calculated as $f_{max} = \sqrt{g/(2\pi h n)}$, where $n = 1/kh \approx 1$. By replacing $h$ with the tidally induced change in water depth $\mathbf{h_{tide}}(t)$, the temporal variations in the theoretical maximum frequency $\mathbf{f_{max}}(t)$ at channels 1253–1256 and 4068–4071 (dashed black line in Fig. 2b, d) agree well with the observed spectral peaks of the HFMN. Conversely, the HFMN and tidal modulation become almost invisible along relatively deep sections of the cable, such as channels 2316–2319 and 5649–5652 (Fig. 2c, e), where the $\mathbf{f_{max}}(t)$ at $h \geq 20$ m would fluctuate below 0.3 Hz (Supplementary Fig. 3) and the cable burial depths are likely deeper due to the topographic depressions (Fig. 2a). However, the passage of Typhoon Muifa is clearly visible on the time–frequency spectrograms of these deep channels. The dominant microseismic noise evolves consistently with the observed wind speed (with correlation coefficient reaching about 0.89 around 0.20 Hz; Supplementary Fig. 4) and typhoon track. The typhoon passage over the

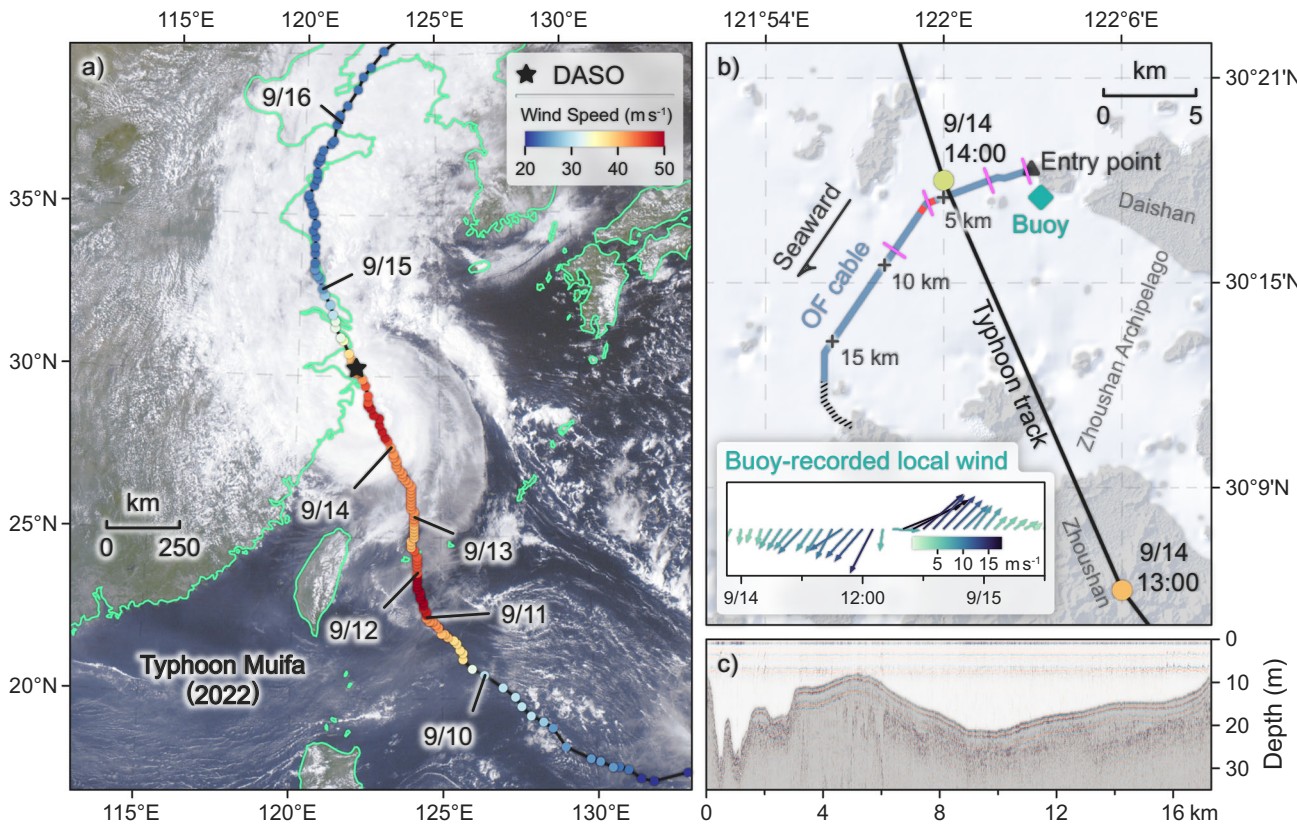

**Fig. 1 | Distributed acoustic sensing observatory (DASO) and track of Typhoon Muifa (2022). a** Regional map showing the typhoon track (curved black line), with the solid color-coded circles representing the wind intensity. The black star indicates the DASO location. The background image is an FY4A/AGRI satellite image of Typhoon Muifa at 06:15 UTC on September 14, 2022. **b** Local map showing the relative locations of the submarine optical-fiber (OF) cable (solid blue line with black crosses at a 5-km interval and dashed black line) and typhoon track (curved black line). The black triangle denotes the entry point of the cable into the sea.

The purple short lines mark the channel locations corresponding with the spectrograms in Fig. 2b–e. The two red segments along the cable are selected for the horizontal current measurements. The nearby ocean buoy is marked by a green diamond, and the corresponding local wind record during the passage of Muifa is shown as colored feather plot. Typhoon Muifa passed over the DASO at around 14:00 UTC on September 14, 2022. **c** Water depth along the ocean-bottom cable starting from the entry point.

DASO can be identified with a "quiet" strip on the spectrogram around 14:00 UTC on September 14, 2022 (bounded by the two dashed white lines in Fig. 2c, d).

It is noticeable that although the high-frequency (>0.3 Hz) component seems to lie within the frequency band (~0.1–0.5 Hz) of secondary microseismic noise (SMN), the observed microseismic noise is directly generated by the OSGW seafloor pressure, namely primary microseismic noise (PMN). Traditional observations of microseismic noise on terrestrial seismic networks or OBSs constitute diffuse seismic energy radiated into the far field, whereas the observation is conducted just beneath the water when a typhoon passed overhead in this study. Because the dominant frequency band of typhoon-generated ocean waves in shallow waters can often span over ~0.1–0.35 Hz[33,34], the corresponding near-field PMN can extend beyond 0.3 Hz. In addition, the microseismic noise is observed tidally modulated and sensitive to water depth. This is contradictory to the generation mechanism of SMN[7], whereby the nonlinear interaction between opposing ocean waves induces a depth-independent pressure fluctuations on the seafloor.

## Measurements of ocean wave propagation

Figure 3a–d shows the $f$–$k$ spectra of the 30-min-long datasets for all 8655 channels at four different stages of Typhoon Muifa. The dominant microseismic energy is detected within the ~0.08–0.38 Hz frequency band and $-0.065$–$0.065\,\mathrm{m^{-1}}$ wavenumber range. The corresponding phase velocity ($c = f/k$) is the apparent phase velocity

of the OSGWs propagating across the OF cable[24,25]. The spectral energy in the right quadrant of each $f$–$k$ spectrum with a positive wavenumber, which possesses a positive phase velocity, mainly originates from OSGWs propagating with a northeast component across the cable toward the coast of Daishan Island (landward); conversely, the spectral energy in the left quadrant, which possesses a negative phase velocity, is generated by OSGWs propagating with a southwest component (seaward). The dominant propagation direction of the OSGWs reverses as Typhoon Muifa passes over the DASO (Fig. 3b, c), mainly because the winds rotate counterclockwise around the typhoon center in the Northern Hemisphere, resulting in predominantly northeast winds and southwest winds blowing over the water when Typhoon Muifa is approaching and departing the DASO, respectively.

Figure 3e–h shows the derived apparent phase velocities and corresponding spectral energies along the entire cable during the passage of Typhoon Muifa, which were calculated via a sliding-window $f$–$k$ analysis of the entire DASO dataset (see "Methods" for details). Seaward OSGWs with apparent phase velocities of ~8–12 m s⁻¹ in the 0.12–0.28 Hz frequency band are prominent along most of the cable before 14:00 UTC on September 14 (Fig. 3e, f); landward OSGWs with similar apparent phase velocities in the same frequency band become prominent after the passage of Typhoon Muifa (Fig. 3g, h). This reversal of the dominant OSGW propagation direction is consistent with the observed changes in sea-surface wind direction at the nearby ocean buoy (Fig. 1b).

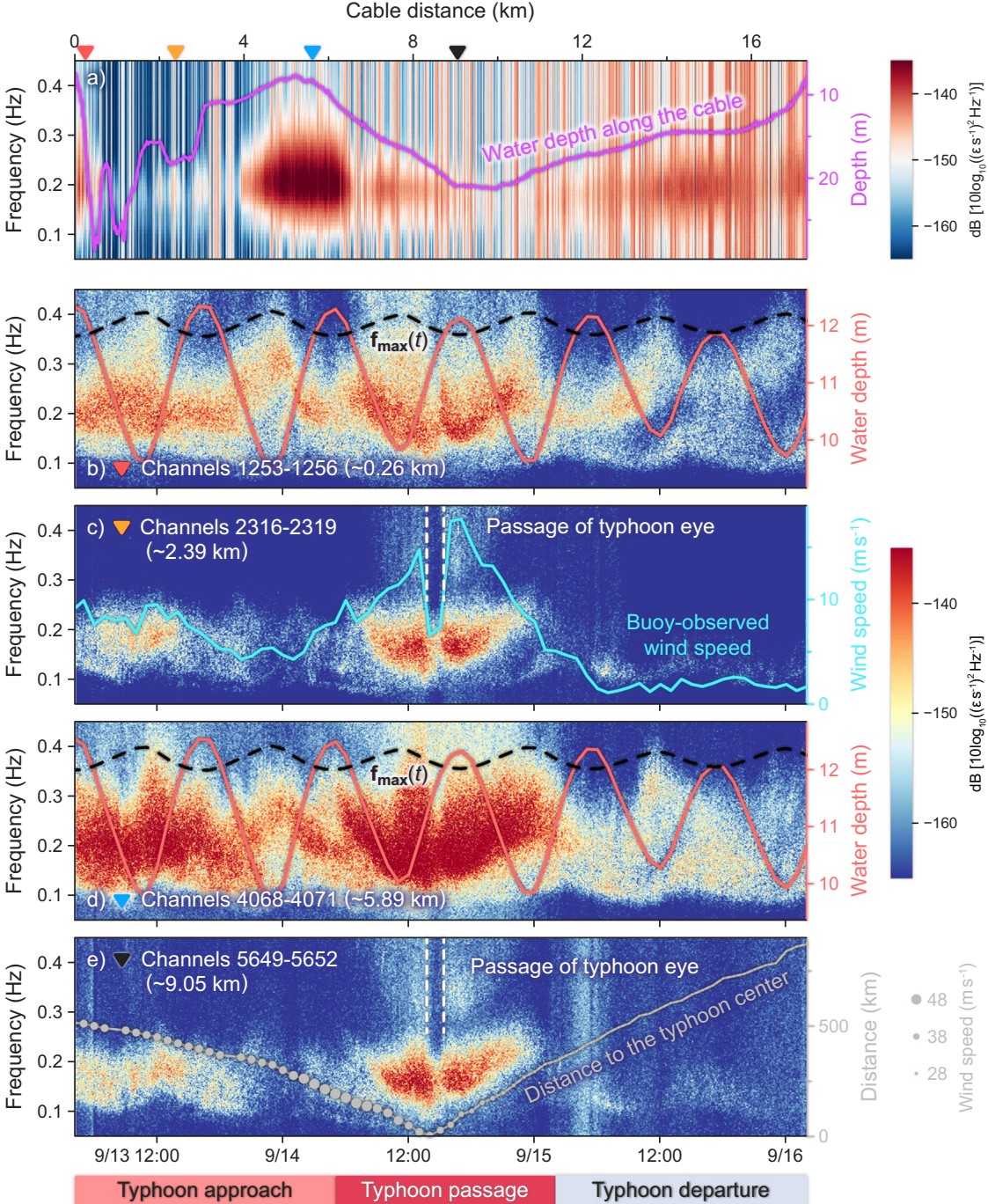

**Fig. 2 | Spatial and temporal characteristics of the microseismic noise recorded by the distributed acoustic sensing observotary (DASO) during the passage of Typhoon Muifa. a** Distance–frequency spectrogram of the upper-quartile spectral power at each channel along the cable, with the corresponding water depth (magenta line) overlain. **b** Mean time–frequency spectrogram of the DASO data along channels 1253–1256 (marked by the inverted coral triangle in (**a**)) for the period from 04:00 UTC on September 13 to 02:00 UTC on September 16, 2022. The overlaid solid coral line represents the tide-modulated water depth $\mathbf{h}_{\mathbf{tide}}(t)$ over channels 1253–1256. The dashed black line represents the estimated maximum frequency $\mathbf{f}_{\mathbf{max}}(t)$ of the seafloor pressure induced by ocean surface gravity waves at channels 1253–1256. **c**–**e** Same as (**b**), but for channels 2316–2319, 4068–4071 and 5649–5652 (marked by the orange, blue and black inverted triangles in (**a**) respectively). The solid cyan line in (**c**) denotes the wind speed observed by the nearby ocean buoy. The solid coral line and dashed black line in (**d**) represent the same as those in (**b**) but for channels 4068–4071. The curved gray line in (**e**) indicates the distance from the centroid of the DASO to the typhoon center, with sizes of the solid circles representing the wind intensity. The two dashed white lines in (**c**) and (**e**) indicate the time period when Typhoon Muifa passed over the DASO.

Previous studies[25,26,28] have indicated that the most prominent feature of the DAS $f$–$k$ spectrum is that the lower edges of the spectral energy packets mainly fit the linear OSGW dispersion curve[35]:

$$f^2 = \frac{gk}{2\pi} \cdot \tanh(2\pi kh) \quad (1)$$

Given that the DAS-observed OSGW propagations are apparent, these lower edges are typically thought to be caused by OSGWs propagating axially or paraxially along the cable. However, the fit between the lower edge of the spectral energy packet and the linear OSGW dispersion curve could be affected by several factors, including (1) the directional wave spectrum, (2) the varying bathymetry along the cable,

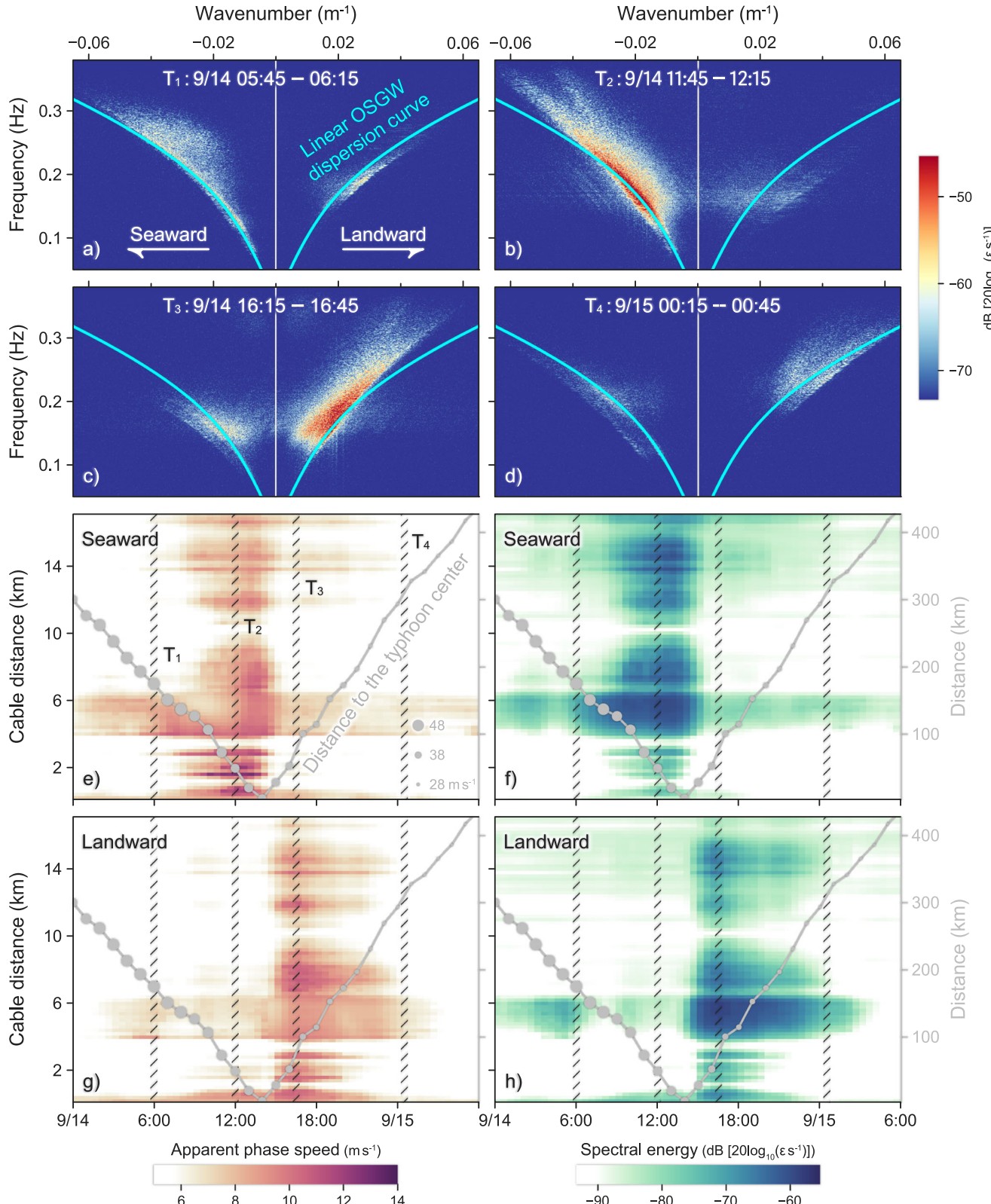

**Fig. 3 | Frequency–wavenumber (*f–k*) spectra of the distributed acoustic sensing observatory (DASO) observation data and inverted ocean surface gravity wave (OSGW) propagation along the entire cable during the passage of Typhoon Muifa.** The *f–k* spectra are calculated at four different stages along the track of Typhoon Muifa (passes over the DASO at -14:00 UTC on September 14, 2022): (**a**) ~06:00 UTC on September 14; (**b**) ~12:00 UTC on September 14; (**c**) ~16:00 UTC on September 14; and (**d**) ~00:30 UTC on

September 15. The overlaid solid cyan lines represent the linear OSGW dispersion curves for each stage and an assumed 12.5-m water depth. Spatio-temporal variations in (**e**), (**g**) the seaward- and landward-propagating phase speed and (**f**), (**h**) corresponding spectral energy along the entire cable. The curved gray line indicates the distance from the typhoon center to the DASO, with the size of the solid circle representing the wind intensity.

and (3) the ocean current fields above the cable. Considering directional wave spectra with little energy propagating axially or paraxially down the cable, all OSGWs propagate obliquely across the cable and produce apparent phase velocities faster than those predicted by the linear OSGW dispersion relation. On the other hand, the nonflat cable layout would distort the observed apparent OSGW wavenumbers and therefore the apparent phase velocities. It may also lead to multiple dispersive spectral energy packets with lower edges corresponding with OSGW dispersion curves of diverse water depths, as exemplified in Fig. 3a–d. And OSGWs shoaling-up over a slope bathymetry would become higher and steeper, resulting in nonlinear amplitude dispersion effects on the wave phase speeds[36,37]. Additionally, the underlying ocean currents would induce Doppler effects into the OSGW dispersion relationship[38–41], which would manifest as non-reciprocal frequency shifts of the spectral energy packets in the DAS $f$–$k$ spectrum[26] (Fig. 4b, c). Nevertheless, the Doppler effect can also be used to measure the ocean current.

Ocean current measurements, based on Doppler shifts of OSGW fields, have been successfully conducted using radar and optical systems[39,40,42–44]. These measurements can characterize the spatial variations of currents, yet they are reliant on the weather. The DAS-based observations of the current-induced Doppler-shifts, on the other hand, have been recently proven effective for measuring current speed[28]. These methods, which use the OSGW-generated pressure disturbances at the seafloor, can function effectively even in harsh weather conditions. It is anticipated to offer a more cost-effective means of measuring submesoscale currents than ocean acoustic tomography[45,46] and conventional oceanographic instrumentation, such as moored current meters and acoustic Doppler current profilers. In the following section, an approach utilizing the DAS $f$–$k$ spectra to characterize ocean currents is proposed. More importantly, even during typhoon conditions, this method can estimate current direction in addition to measuring current speed.

## Ocean-current measurements

In this study, the horizontal ocean current flowing over the DAS-instrumented cable is successfully measured using a proposed method (see Methods for details). For example, Fig. 4b, c show the measurement results of the mean ocean current during the period 16:40–16:50 UTC on September 13, 2022, using two adjacent segments (channels 3875–4075 and 4225–4425; red line segments in Figs. 1b, 4a). The OSGW dispersion curves for both cable segments are obviously affected by the current-induced non-reciprocal Doppler shift, which is apparent as a clockwise rotation of the dominant spectral energy packets with respect to the linear OSGW dispersion curves (cyan curves, calculated via Eq. (1)). A synchronous fit of the Doppler-shifted OSGW dispersion curves (black curves, calculated via Eq. (3) in "Methods") to the lower edges of the dominant spectral energy packets along both segments yields respective mean ocean-current speed and direction of -0.7 m s$^{-1}$ and ~240°.

Figure 4d shows the ocean current flowing over the two selected cable segments during the passage of Typhoon Muifa at a 10-min resolution. A video of the stepwise OSGW dispersion curve fit and corresponding ocean-current measurements is provided (Supplementary Movie). The temporal variations in the measured ocean current are mainly consistent with the tidally induced sea-level fluctuations, with the changes in speed and direction dominated by the tidal current. Similarly, the M2 tidal current has been measured using a DAS-instrumented OF cable[28]. However, the measured ocean current reverses and strengthens irregularly during the passage of Typhoon Muifa (~11:00 to 19:00 UTC on September 14). This is consistent with the observed sea-surface wind fluctuations at the nearby ocean buoy (Fig. 4e). The correlation coefficient between the measured ocean current directions and observed wind directions is 0.902 (Supplementary Fig. 5). Overall, the above results

demonstrate the feasibility of estimating ocean currents (speed and direction) in a shallow water under typhoon conditions via our proposed DAS-based method.

We note that there are potential improvements to our DAS-based ocean-current measurement method. Firstly, despite the endeavors undertaken to sharpen the lower edges of dominant energy packets in $f$–$k$ spectra for more accurate fitting (see details in Methods), it should be acknowledged that human visual perception in the manual fitting process is intrinsically prone to bias, particularly when dealing with dispersion curves with slight Doppler shifts in a smeared $f$–$k$ spectrum image. Therefore, an algorithm-based automated fitting process, such as one based on digital image processing or machine learning for edge detection and extraction, is crucial to improve the efficiency and accuracy in follow-up work. Secondly, the nonlinearity of OSGWs would bias the ocean current measurements based on this linear-theory-based method. For example, the nonlinear amplitude dispersion effects of OSGWs with high wave steepness ($k\zeta$) over varying shallow bathymetry would lead to higher OSGW phase speeds than those predicted by the linear dispersion relation[37,47]. So adoption of a proper nonlinear OSGW dispersion relation (e.g., nonlinear Boussinesq theory[48]) and simultaneous deployment of wave height sensors and tide gauges are expected to alleviate the nonlinear effects. Lastly, it is recommended to install cables that are specifically designed for the purpose of observing ocean currents (e.g., Supplementary Fig. 6a, b), instead of employing pre-existing undersea cables that are rigidly regulated and predetermined in their layout. For example, horizontal arrangement of the cable segments as shown in Supplementary Fig. 6a can effectively mitigate the impact of varying bathymetries discussed in the previous section. Schemes of cable layouts shown in Supplementary Fig. 6b are designed to improve the spatial coverage and resolution of ocean current measurements.

It is important to emphasize that, despite its intrinsic limitations, this approach is expected to have potential implications for oceanography. Given the fact that OSGW-induced seafloor pressure perturbations decay exponentially with water depth, the practical implementation depth $H$ of this method is theoretically capped at one-half of the OSGW wavelength, i.e., $H \le \pi k^{-1}$. The $H$ is also related to the normalized horizontal compliance $\eta_x$ of the DAS system, which is the conversion coefficient between the $\mathbf{\Delta P}$ and horizontal strain rate and affected by various factors[28] (e.g., cable-seafloor coupling and cable construction). The $H$ would be deeper for unburied high-sensitivity OF cables than those for telecommunications and power cables used in related studies (e.g., $H \approx 100$ and 150 m reported by Sladen et al.[25] and Williams at al.[28], respectively). Therefore, this approach could be potentially used to monitor tidal currents and issue prior warning for extreme tide events (e.g., storm surges) in coastal areas. In addition, this method measures the depth-averaged horizontal current from ocean surface to the seafloor or the unburied cable. Therefore, measurement of non-uniform 3D currents by this method would be prone to bias (e.g., the biased measurement of linear shear currents exemplified in Supplementary Fig. 7). Yet, this method may have potential for observations of ocean motions with significant vertical components (e.g., submesoscale eddies, internal waves) by utilizing specially planned OF cables. For example, a scheme of multi-layer cable layout as shown in Supplementary Fig. 6c allows for estimation of vertical variations of ocean currents by synchronously measuring the depth-averaged horizontal currents at different layers. It is anticipated that the DAS-based techniques would further facilitate observations of oceanographic phenomena, including tide and internal waves[30], and upwellings/downwellings essential for the marine ranching.

## Methods
### Data processing
The raw DAS dataset **DATA$_{raw}$** with the unit of phase change rate (rad s$^{-1}$) is first converted to dataset **DATA$_{strain\ rate}$** with the unit of

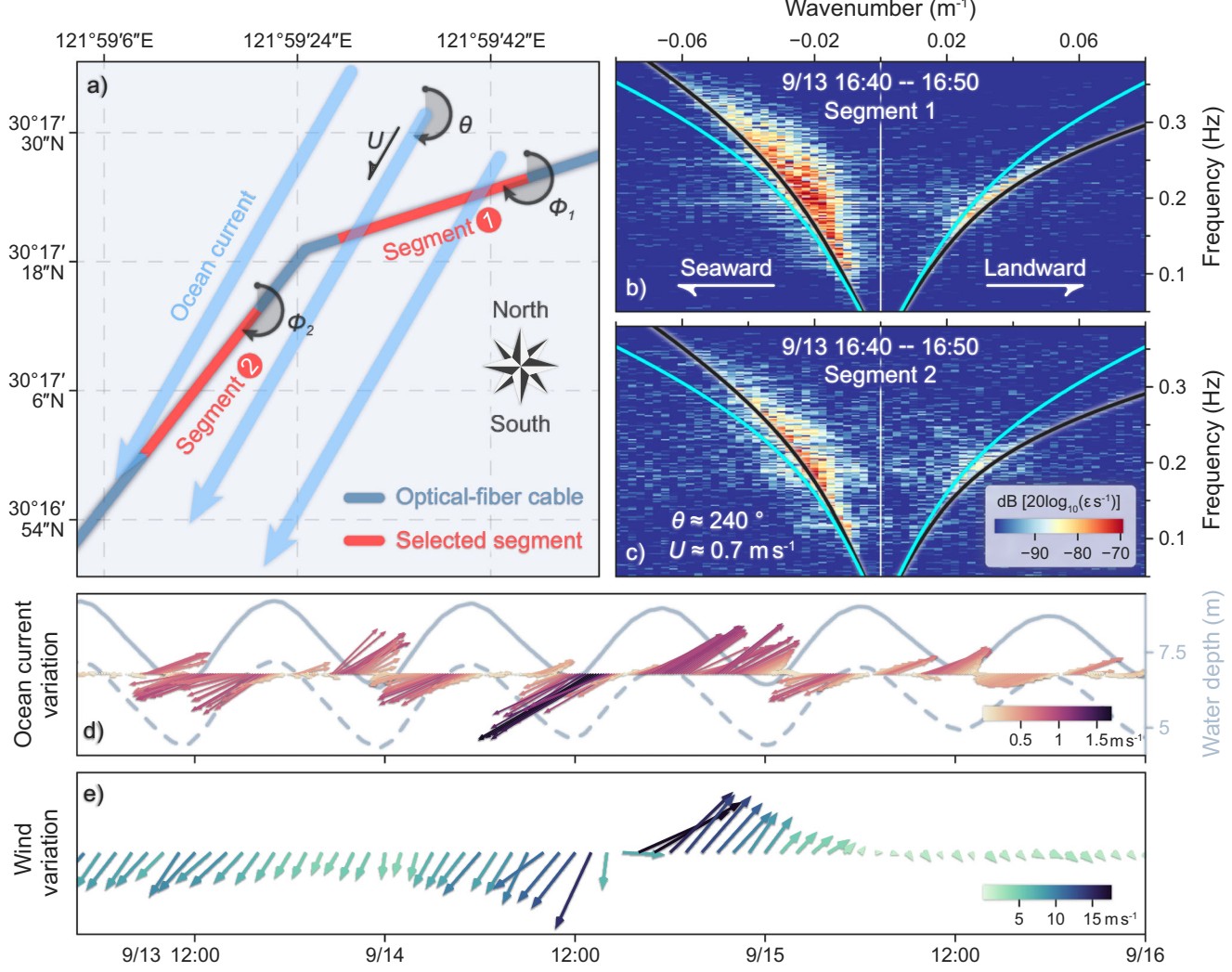

**Fig. 4 | Ocean-current measurements during the passage of Typhoon Muifa.**
**a** Schematic illustration of the ocean-current measurements based on distributed acoustic sensing. Two adjacent segments with varying azimuths $\Phi$ (solid red lines; Segment 1: channels 3875–4075, $\Phi_1 = 249.7°$; Segment 2: channels 4225–4425, $\Phi_2 = 214.3°$) along the optical-fiber cable (solid dark-blue line) are selected to measure the horizontal ocean current (pale blue arrows) flowing over the cable. The frequency-wavenumber spectra are calculated using the 10-min observations along (**b**) segment 1 and (**c**) segment 2 during 16:40–16:50 UTC on September 13, 2022.

The overlaid solid cyan lines represent the linear ocean surface gravity wave dispersion curve, and the solid black lines denote the dispersion curves under the influence of an ocean current that is propagating at -0.7 m s$^{-1}$ and $\theta = \sim 240°$. **d** Feather plot of temporal variations in measured ocean-current speed at 10-min intervals. The dashed and solid gray lines represent the tide-modulated water depths $h'_{1,2}(t)$ for segments 1 and 2, respectively. **e** Feather plot of temporal variations in the observed sea-surface winds at the nearby ocean buoy.

strain rate ($\varepsilon$ s$^{-1}$) as follows:

$$\text{DATA}_{\text{strain rate}} = \frac{\lambda}{4\pi n G \xi} \cdot \text{DATA}_{\text{raw}}, \tag{2}$$

where $\lambda = 1.550 \times 10^{-6}$ m is the laser wavelength of the DAS system, $n = 1.467$ is refraction index of the OF, $G = 2$ m is the gauge length and $\xi = 0.78$ is the photoelastic scaling factor for longitudinal strain. The **DATA**$_{\text{strain rate}}$ is then processed using the Python package Obspy, including: (1) demeaning and detrending; (2) decimating to 2 point per second and 1-Hz low-pass filtering. Noted that the DAS data used in $f$–$k$ analysis is further decimated to 1 point per second for calculation efficiency.

### Sliding-window f–k analysis

A sliding-window $f$–$k$ analysis is applied to the whole DASO-recorded dataset to derive the apparent phase velocity along the entire cable during the passage of Typhoon Muifa. The sliding-window is two-dimensional, and the window lengths are set to 1 h and 400 m in the time and space dimensions, respectively, both with 50% overlap. Each windowed dataset is then transformed to a $f$–$k$ spectrum via a 2D Fourier transform. The dominant apparent phase velocity of each $f$–$k$ spectrum is further calculated by averaging the apparent phase velocities at the $f$–$k$ bins where the spectral energy is greater than 60% of the maximum spectral energy of each $f$–$k$ spectrum. The spatio-temporal variation of the apparent phase velocity is finally derived on the basis of all of the dominant apparent phase velocities.

### Ocean-current measurements from DAS-instrumented cable observations

Figure 4a provides a schematic illustration of the proposed ocean-current measurement method. Assuming that a uniform horizontal ocean current flows over two adjacent segments with different orientations, the current speed $U$ and direction $\theta$ can be estimated by synchronously fitting the Doppler-shifted OSGW dispersion curves to the lower edges of dominant spectral energy packets in the $f$–$k$ spectra of

both cable segments. The Doppler-shifted OSGW dispersion curves are expressed as follows[26]:

$$\begin{cases} \left(f_1 + \frac{U}{\cos(\theta - \Phi_1)}k_1\right)^2 = \frac{gk_1}{2\pi} \cdot \tanh\left(2\pi k_1 h_1\right) \\ \left(f_2 + \frac{U}{\cos(\theta - \Phi_2)}k_2\right)^2 = \frac{gk_2}{2\pi} \cdot \tanh\left(2\pi k_2 h_2\right) \end{cases}, \Phi_1 \neq \Phi_2 \quad (3)$$

where $\Phi_{1,2}$ and $h_{1,2}$ represent the azimuth angles and water depths of the two selected cable segments, respectively, and $f_{1,2}$ and $k_{1,2}$ are the frequencies and apparent wavenumbers of the two corresponding $f-k$ spectra, respectively. $\frac{U}{\cos(\theta - \Phi_{1,2})}k_{1,2}$ is the Doppler shift term induced by the ocean current. The cable segment selection criterion is that the two adjacent segments should orient differently but within the prevailing wave direction range under stable sea states.

The specific calculation details for this study are described as follows. The cable channels 3875–4075 and 4225–4425 (at 5.5–5.9 km and 6.2–6.6 km respectively; red line segments in Figs. 1b, 4a) are selected according to the above selection criterion and their relatively higher signal-to-noise ratios (Fig. 2a, d). Both selected cable segments consist of 200 channels and possess respective azimuths $\Phi_1 = 249.7°$ and $\Phi_2 = 214.3°$ (Fig. 4a). The DAS observations along each cable segment are transformed into a series of consecutive $f-k$ spectra with both the sliding-window length and step equal to 10 min. The $f-k$ spectra for both cable segments at each time window are then paired for the fitting process. We have developed a dedicated graphical user interface (GUI; Supplementary Fig. 8) based on the Python packages Numpy and Matplotlib for manual fitting process. In the case of given $\Phi_{1,2}, h_{1,2}, f_{1,2}$ and $k_{1,2}$, this GUI allows for the manual adjustment of the "Current speed" and "Current direction" Sliders to conduct synchronous fitting between the corresponding Doppler-shifted OSGW dispersion curves and the lower edges of the dominant spectral energy packets in the $f-k$ spectra of both cable segments. The water depths $h_{1,2}$ in Eq. (3) are replaced by the tide-modulated water depths $\mathbf{h}'_{1,2}(t)$ (Fig. 4d), which are estimated as follows:

$$\mathbf{h}'_{1,2}(t) = D_{1,2} + \mathbf{T}(t), \quad (4)$$

where $\mathbf{T}(t)$ is the tide level variation of the DASO region, with temporal resolution of 10 min after linear interpolation. Due to the lack of exact information of cable layout and burial, the characteristic water depths of the two selected cable segments $D_{1,2}$ are estimated using a specialized GUI (Supplementary Fig. 9). Based on the $f-k$ spectra with inconspicuous Doppler shifts of cable segment $x$ ($x = 1\ or\ 2$) and known $\mathbf{T}(t)$, this GUI allows for manual adjustment of the "$D_x$" Slider to control the variation of $\mathbf{h}'_x(t)$ to optimize the fit between the corresponding linear OSGW dispersion curves and the lower edges of dominant spectral energy packets. In both GUIs, the dominant spectral energy packets in all of the $f-k$ spectra are highlighted by uniformly rendering with a diverging colormap and filtering out the spectral energy below −115 dB.

## Data availability
The FY4A/AGRI satellite image is provided by the National Satellite Meteorological Center, China Meteorological Administration from their website http://satellite.nsmc.org.cn/portalsite/default.aspx. The tide level data used in this work is available from the National Marine Data Center at http://mds.nmdis.org.cn/pages/tidalCurrent.html. The typhoon track data is available at http://typhoon.nmc.cn. The spectra data displayed in Figs. 2 and 3, and 1 Hz DAS data used for current measurements in this study is available on a public data repository[49]. The whole datasets of DAS, seismic profile and ocean buoy observations are available from the corresponding author on request.

## Code availability
The two specialized GUIs used for current measurements in this study is available on a public data repository[49]. The code and scripts used to analyse the data and to generate the plots in this paper are available from the corresponding author on request.

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

## Acknowledgements

This work was supported by the Key Research and Development Plan of Zhejiang, China [grant number 2021C03181, J.M.L.], the National Natural Science Foundation of China [grant number 61831020, W.X.; 42274067, J.M.L.], the Science Foundation of Donghai Laboratory [grant number DH-2022ZY0001, J.M.L.], the National Key R&D Program of China [grant number 2022YFC3102202, B.S.W.], and the Chinese Academy of Sciences (CAS) Project for Young Scientists in Basic Research [grant number YSBR-020, B.S.W.]. We thank Smart Earth Sensing (Hefei) Technology Co., Ltd for providing the DAS instrument and assistance during the experiment, and Zhoushan Branch of China Mobile Zhejiang Co., Ltd for allowing us to use the dark fiber within their submarine cable.

## Author contributions

J.M.L. wrote the manuscript and proposed the central idea. S.K.F. conducted the analysis under J.M.L.'s instruction and contributed to improving the manuscript. R.J.H., Q.S.T. and F.Z.Q. were involved in the field experiment and interpreting the results. B.S.W. guided the whole experiment and supported remotely DAS operation. W.X. supported the initial idea and gave strong input in the final version of the manuscript.

## Competing interests

The authors declare no competing interests.
