## [Peer Review File · Nature Communications]

Monitoring ocean currents during the passage of Typhoon Muifa using optical-fiber distributed acoustic sensingREVIEWER COMMENTS

Reviewer #1 (Remarks to the Author):

Key results:

- Ocean current speeds and azimuth are derived from a method based on relative current-induced Doppler shifts on shallow, contiguous segments of a DAS cable having different orientations
- The measured ocean currents are consistent with tidally induced sea-level fluctuations, typhoon-induced currents and sea-surface winds observed at a regional ocean buoy
- The novel detection and visualization of the wind-generated current structure of a major cyclone by a DAS system underneath

Validity: The methodology proposed is valid and useful in various contexts. However, some clarifications of its implementation are required, while a detailed discussion of the results is considerably lacking. A few conceptual shortfalls/gaps are present that have to be addressed, but these do not affect the validity of the results in general.

Originality and significance: The work is original and of immediate interest for the community, notably physical oceanography, ocean engineering, and other ocean-related fields, geophysics and fiber-optic sensing in general.

Data & methodology: The theoretical framework of the approach is valid. The data has enough quality to identify the physical signals of interest. However, the method is poorly explained and there appears to exist some degree of ambiguity in the workflow that might compromise the reproduction of the results. Notably, the presentation of requirements, validity ranges and limitations of the method is unclear or insufficient. Although some of the assumptions of the method are mentioned, these could be stressed some more while others are missing. Some literature is missing or was not acknowledged properly. For specifics, I've listed the major comments below.

Appropriate use of statistics and treatment of uncertainties: Yes, up to some missing descriptions of the correlation methodology implemented.

Conclusions: The conclusions and data interpretation are mostly valid, although perhaps not sufficiently backed-up by more detailed descriptions and discussions, which would in turn facilitate an assessment of the robustness and reliability of the method.

Major comments:

1. The key fact that only the horizontal component of the currents is retrieved (i.e. horizontal currents are assumed) is not mentioned anywhere in the text and should be emphasized in the abstract and in at least a couple sections more. On the other hand, a discussion on the detection potential of the method for ocean circulation structures with important vertical components (such as eddies, internal waves or down-/upwelling) is missing. For example, could this method discern between deep/surface currents? or between currents and other (sub)mesoscale anomalies?
2. Details on the manual picking and fitting process of the doppler-shifted linear dispersion curves implemented and its limitation are lacking. This is quite relevant, as the parameterization of the f-k transformation (from which the dispersion metrics are obtained) plays a major role on its SNR and its time-space resolution trade-off. As the method strongly relies on some assumptions, namely that: background currents are horizontal; locally steady and uniform; and that the segment layouts are flat (i.e. constant depths), the authors are requested to thoroughly discuss potential biasing induced by non-uniform 3D currents and sloping/complex bathymetry. Additionally, as the exact cable layout at the seafloor is often unknown, the recommendation for proper georeferencing/surveying of the segments should be stressed. Also, some comments on potential effects related to the spatial stability/variability of the directional wave spectrum between the segments would make sense.
3. A key point stressed by the authors is that the analysed signals correspond to microseisms. As the frequency band of the most energetic signals considered is between 0.1-0.3 Hz, which is well-known to lie within the secondary microseismic (double-frequency) band, does this imply that all the phenomena observed are caused by non-linear wave-wave interactions? On the other hand, this same frequency band corresponds to the dominant surface gravity wave peak in shallow areas and, as the authors mention, the estimated phase speeds are between 8~12 m/s, which are way too slow for microseisms while are at the same time consistent with typical phase speeds of ocean surface gravity waves. It therefore seems more plausible that all the signals analysed by the authors are being induced directly upon the cable by gravity waves.
4. L 191-6: Although the slowest apparent component is more likely to correspond to the true wavefield speed, this is not always necessarily the case. As an example, what if several swells with different speeds are superposed in a way that those that arrive (nearly) axially are even faster than those at an angle? This is particularly relevant for cyclones, as their directional wave spectrum can be complex and asymmetric among its four quadrants (see e.g. Hu & Chen, 2011-Directional spectra of hurricane-generated waves in the Gulf of Mexico). The fact that previous studies have fitted linear gravity wave dispersion curves to their f-k spectra does not imply that the observed waves propagated parallel to the cable, as f-k dispersion branches could still be apparent. Assuming true propagation along a line is generally incorrect and may generally lead to errors, as only non-axial components might occur. However, the fact that all observed wavelengths are apparent does not appear to affect the results, but the fact should be clarified, specially for the non-specialist readers.
5. L 255-7 The authors mention that the picking and fitting is done manually. Although the supplementary video showing the several picked f-k spectra are helpful to visualize this, further details on the workflow and criteria used to perform this picking and fitting are necessary. Also, do the authors envision a potential way to effectively and accurately automatize the procedure?
6. L 258-60 The selection criteria of the three channel ranges shown in Fig 2 was not entirely clear to me

until I read these lines. The fact that these kind of specific cable segments are required for the method to work should be mentioned somewhere in the introduction, at least. Furthermore, it is somewhat confusing that none of the data segments highlighted in Figs. 1 and 4 is studied in Fig. 2. It would make sense to at least highlight the segments in Fig. 2.

7. It is known that the detection of surface gravity waves with seafloor DAS is not possible at all depths, as these can attenuate significantly over deep waters, as implied by the eq. referenced in L 138. This poses a major, additional limitation to the method, as it could only be implemented over sections that are not too deep. As far as I see, this limitation is not mentioned and should be clearly stated. For instance, how does the reliability of the method vanishes as a function of depth/wavelength/waveheight? and what are the useful cable depth ranges for the method to work?

8. L 305 – Some parts of the methods are not explained in much detail, such that several questions are left open. For instance, What is the selection criteria of the segments? Also, how exactly is the fitting process performed to solve for the system of equations (2)? If, as stated in L324-5, energy below 17dB is filtered for every window, does this imply that the minimum speed dispersion curve envelopes may not be picked or are not being sought at all? If picking is done manually, doesn't the colormap scale affect the interpreter's choice? Additionally, details about the uncertainty of the method are missing. This may include a discussion on error propagation during the fitting process, which is prone to ambiguity for noisy/smearred fk images.

9. L 313 – The length of each of the selected cable segments is about 400 m (according to the channel spacing reported of 2m), but a constant depth is assigned to each. Is the bathymetry truly flat for these segments? If not (as it appears from the bottom profile), how are these constant depth values defined? and how exactly does the assumption of flat bathymetry under a sloping segment affect the results? This could be e.g. quantified/shown in a figure.

10. A data processing section is missing. Any preprocessing of the data (e.g. unit conversion, filtering, decimation, detrending, etc.) prior to the implementation of the method (or the lack of any processing step) must be reported accordingly.

Suggested improvements:

- As the study deals specifically with the estimation of the magnitude (speed) and azimuth of ocean currents, the proposed title might be a bit too broad and ambiguous. My suggestion would be to change it to "Directional (or azimuthal) monitoring of ocean currents during the passage of Typhoon Muifa (2022) using optical-fiber distributed acoustic sensing" or something similar that emphasizes ocean currents. As there is a recent study on deep sea current detection with DAS using a different approach (hanging cable sections) by Mata Flores et al. (2023), the suggested title would emphasize the fact that your method additionally retrieves current azimuth, as opposed to the former study, that only retrieve magnitudes. Also, including a brief comparison of your method with that of the former study seems appropriate.

- It would encourage the authors to discuss potential hydrodynamic effects on the measurements. For instance, how is the method expected to be affected by the potentially non-linear (high $k\zeta$) wave regime of large storms (such as a typhoons) where linear theory no longer applies and waves can break and interact?

- In Fig 1b, the spatial extent and wind direction of the cyclone would be useful to understand the

expected wave directional distribution during the typhoon during closest approach (e.g. contours indicating the extent of the maximum wind speeds/ROCI/isobars/...). Are there any models of wave/wind stress direction available for the region, such as hindcasts?

- In Fig 2 the authors could use different colors (not repeating them) for each of the curves overlain to improve clarity of the figure. Also, reporting the depths of the shown channel ranges in the figure would help the reader. Including a fourth spectrogram panel for some of the shallow channels around ~5km along the cable (e.g. those used in Fig. 4) might be interesting, as the spectral power seems quite high here.
- In general, it would be interesting to see some more discussion on the physical oceanographic implications and applications of the method, at least in a general way. This could optionally focus on the regional oceanographic setting (e.g. well-known surface/deepsea circulations in the region) where the experiment was carried out. For instance, is it possible to link the estimated current speed and azimuth values to previous observations in the region or observations for other cyclones? It is remarkable that tidal modulations and tidal currents can be observed so clearly in the data. As the typhoon-induced currents seem to be canceled or even reversed by the tides, could this mean that some currents might be overshadowed by the local tides if not fast enough? The method could also potentially be used to retrieve tidal components under steady conditions, a fact that is worth highlighting.
- This and other DAS methods have a potentially high impact for oceanography, so that cable dedicated deployments might be planned in the future. What would the author's suggestions be for future studies planning to lay cables targeting ocean currents? For example, in terms of geometry, location, etc? The authors could also highlight that the simultaneous use of multiple cables or additional sensors (e.g. current-meters, buoys, moorings, sensor arrays) might provide a complementary view of the oceanographic regime.

References:

- L53-65 – Although some references were included, in my opinion, it would be fair to add a few more on cyclone-related microseisms, considering the vast amount of literature available, including review papers and books.
- L60-5 – The cited reference of Davy et al., 2014 is very relevant for the current work as it dealt with the detection of a cyclone passing overhead an array of broadband OBSs and hydrophones in the ocean, in analogy with the presented case of typhoon Muifa with DAS. However the way that this work is referenced and discussed in the text does not seem make justice to this fact (which is not highlighted at all) nor to its findings in general. For instance, in L64 I find “limited” far more accurate than “prohibited” (see Sec. 6 in Davy et al, 2014). At the same time, I disagree that it was only the sparsity of stations in that work what limited the storm tracking (which in fact I consider quite remarkable given the variable geometry, areal extent and the complexity of the gravity wave and bathymetry-dependent seismic noise radiation patterns of a cyclone, specially in the near-field) and I also disagree that this “...prohibited... a detailed near-field investigations of the TC”, as the authors of that work showed in detail and with good data quality the wave polarization and amplitude attenuation functions induced by the cyclone passing overhead, amongst others.
- L66-84. A few references could be added that are relevant for surface gravity wave analyses with DAS: Guerin et al., 2022 and Sladen et al., 2019. For ocean microseism studies with DAS: Xiao et al, 2022 and

Tonegawa et al., 2022. On the other hand, are there any previous studies addressing surface gravity waves and their relationship with ocean currents from theoretical perspectives or proposed experimental methods not related to DAS, such as acoustic/remote sensing/oceanography? This would be worthy to be mentioned and discussed for general context.

- L 212-14 – this is a quite well-known fact within the oceanographic community, thus adding a few older references would be very pertinent e.g. Leckler, 2015 - Analysis and Interpretation of Frequency–Wavenumber Spectra of Young Wind Waves

Clarity and context:

- L20 - “...current-induced Doppler shifts”: the authors could indicate what particular physical signal experiences this effect
- L33 - “Tropical cyclones (TCs)”: Perhaps this study could be safely generalized for Cyclones?(this would implicitly also include subtropical and extratropical)
- L49 - “Microseismic noise (~0.05–0.5 Hz)...”: Ocean microseismic noise (as e.g. lakes can generate microseisms at up to 2 Hz)
- L73 – small typo: “optical-fiber”
- L87-9 - “...These observations show that the ocean-bottom DAS-instrumented cable is sensitive to typhoon-induced microseismic noise, even at ultra-low frequencies (<0.1 Hz), and effective in deriving information on OSGW.”. How do the authors tell apart secondary microseisms from SGWs in the frequency range of interest?
- L 108: is the buoy really collocated with the studied cable segments? If not (as it seems), what is the inter-distance for each case?
- L 165: in Fig 2a was the data lowpassed at 0.4Hz? The decrease above this bound seems abnormally sudden
- L 146. The same phenomenon was reported and studied in detail in ref #22.
- L 147-49. Since $kh \sim 1$ seems to be a constrain of the cited equation, is it then only valid for deep waves? Fig 2b is convincing but the authors could also include the same frequency limits for Figs 2c,d
- L 158: The authors could comment somewhere on the specific correlation method used to obtain the coefficient. The same applies to figures S1 and S2.
- L 183. “with a northeast component” would be more accurate. A similar comment applies to L185
- L 186-190: As the typhoon approaches from the south, westward winds occur just before its arrival, while eastward winds dominate afterwards, which is contrary to what is written in the text. This can be confirmed by the dominant seaward energy in Fig 3b and the opposite in 3c.
- L198-200: I fail to understand the exact connection of this sentence to the previous ones: the fact DAS can provide measurements of surface gravity waves has already been demonstrated.
- L211 – what other factors?
- L 252-3 – what are the specific requirements for the method to work on the alluded environments?
- In Figure 3, between ~4-6 km along the cable there is a shallow cable section of high wave energy which also increases as the storm approaches, both seawards and landwards. However, the wave speeds appear to increase a bit more in the seaward component relative to the background level than they do after the storm leaves in the landward component. What could cause this?
- L 312 The wavelengths retrieved from a DAS segment are always apparent, meaning that the

wavenumbers (k_1 , k_2) are apparent too. This should be emphasized, as the method would eventually not allow for the recovery of the propagation angle of the waves. Some additional tests or discussion on the expected performance of the method within the parameter domain would help in its validation, i.e. as a function of the amplitude and angle of incidence of the waves relative to the cable, the relative angle between the segments, the water depth, the strength and azimuth of the current, etc.

- L 319 – How are these tide-modulated depths obtained? Is a reference tide gauge required for the method to work? If so, this should be clearly stated.
- The seismic profile/bathymetry data shown in Fig 1c does not seem to be mentioned in the Data availability statement.
- The gauge length, time sampling rate and unit measured by the DAS system (strain/strain rate?) are missing and should be reported.

Inflammatory material: None

Reviewer #2 (Remarks to the Author):

This is a review of the manuscript "Monitoring the sea state during the passage 1 of Typhoon Muifa (2022) using optical-fiber distributed acoustic sensing". This manuscript presents a very nice observation of the effect on Distributed Acoustic Sensing of a typhoon propagation over a fiber optic cable. This is an exceptional setting to study the seismic signals generated by strong oceanic events. I do not have many comments and I found this study very interesting but I believe a few things should be considered.

While I understand the paper is observational, I think the authors should discuss at least a little the waves that are observed on the signals. There is no real discussion of this while I feel necessary since phase velocities are discussed. One disadvantage of DAS data is that observations are only made over one dimension. Consequently, a discussion of wave propagation and types cannot be avoided in my opinion. The authors show very nice observations but the impact could be discussed more.

I had to go back and forth a lot to use the figures and I think a few changes could be made to help. In Figure 2 b,c,d the locations are defined as channels but in 2 a and Figure 1 the distance is represented in km. Moreover, the water depth is an important feature so I would make that more directly clear for the different cases.

It would be interesting to represent the location of the particular focus zones showed on Figure 2 also in Figure 1. Moreover, the authors show how the tidal signals dominate in shallow water on one side of the fiber. This was quite surprising and interesting for me. I believe it would be useful to show the observations on the other side of the fiber. Indeed, as the slope is very different, a comparison would be very informative.

Minor comments:

- Line 17: I am not sure I would use the term “hydrological” in the first sentence since it suggests that is what DAS data obtains. I am not sure it is there yet.

- Line 33-36: The authors mention the damage on land and need for intensity estimation at the beginning of the paper. To me this implied that there was estimation in the manuscript of intensity on land which I realized later was not the case.

- Line 89 : Frequencies inferior to 0.1 Hz are not “ultra low” in seismology. I would suppress the ultra.

- Figure 2: It could help to add the colored triangles from a) in b,c,d to make the reading of the figure easier

- Line 229-230 : I understand that the authors detail the methodology in the method part but something should still be said about it apart from the fact that it is “novel”.

Point-by-Point Response to Comments by the Editor and Reviewers

NOTE: Line numbers indicated in this file correspond to the revision-tracked manuscript (PDF version) with the changes noted.

Reviewer #1

Key results:

- “Ocean current speeds and azimuth are derived from a method based on relative current-induced Doppler shifts on shallow, contiguous segments of a DAS cable having different orientations”
- “The measured ocean currents are consistent with tidally induced sea-level fluctuations, typhoon-induced currents and sea-surface winds observed at a regional ocean buoy”
- “The novel detection and visualization of the wind-generated current structure of a major cyclone by a DAS system underneath”

Validity:

“The methodology proposed is valid and useful in various contexts. However, some clarifications of its implementation are required, while a detailed discussion of the results is considerably lacking. A few conceptual shortfalls/gaps are present that have to be addressed, but these do not affect the validity of the results in general.”

Originality and significance:

“The work is original and of immediate interest for the community, notably physical oceanography, ocean engineering, and other ocean-related fields, geophysics and fiber-optic sensing in general.”

Data & methodology:

“The theoretical framework of the approach is valid. The data has enough quality to identify the physical signals of interest. However, the method is poorly explained and there appears to exist some degree of ambiguity in the workflow that might compromise the reproduction of the results. Notably, the presentation of requirements, validity ranges and limitations of the method is unclear or insufficient. Although some of the assumptions of the method are mentioned, these could be stressed some more while others are missing. Some literature is missing or was not acknowledged properly. For specifics, I’ve listed the major comments below.”

Appropriate use of statistics and treatment of uncertainties:

“Yes, up to some missing descriptions of the correlation methodology implemented.”

Conclusions:

“The conclusions and data interpretation are mostly valid, although perhaps not sufficiently backed-up by more detailed descriptions and discussions, which would in turn facilitate an assessment of the robustness and reliability of the method.”

THANK YOU. We appreciate very much for the meticulous review. We have incorporated all the comments by Reviewer 1.

Major comments:

1. “The key fact that only the horizontal component of the currents is retrieved (i.e. horizontal currents are assumed) is not mentioned anywhere in the text and should be emphasized in the abstract and in at least a couple sections more. On the other hand, a discussion on the detection potential of the method for ocean circulation structures with important vertical components (such as eddies, internal waves or down-/upwelling) is missing. For example, could this method discern between deep/surface currents? or between currents and other (sub)mesoscale anomalies?”

DONE. Yes. Only the horizontal component of the ocean currents is measured. Following the reviewer’s suggestion, we have revised the manuscript as follows:

Lines 26-28 (in the Abstract)

Modified

“The ocean current is also derived via a novel method, based on measurements of the current-induced Doppler shifts.”

to

“Further, a novel method based on the current-induced Doppler shifts of DAS-recorded OSGW dispersions is used to calculate both speeds and directions of horizontal ocean currents.”

Lines 123-143

Modified

“These observations show that the ocean-bottom DAS-instrumented cable is sensitive to typhoon-induced microseismic noise, even at ultra-low frequencies (<0.1 Hz), and effective in deriving information on OSGW and ocean-current propagation at high spatiotemporal resolutions during such an extreme weather event.”

to

“These observations show that the ocean-bottom DAS-instrumented cable is sensitive to typhoon-induced microseismic noise, even at ultra-low frequencies (<0.1 Hz), and effective in deriving information on OSGW along the cable. We further presented a novel measurement method, which is based on two specific cable

segments with different orientations, to successfully reveal both speeds and directions of the horizontal ocean-current component at high spatiotemporal resolutions during such an extreme weather event.”

Line 182 (in the caption of Figure 1)

Modified “current measurements” to “horizontal current measurements”

Lines 382, 470, 496, 530

Modified “ocean current” to “horizontal ocean current”.

We acknowledge that this method cannot discern between deep/surface currents with a single submarine cable, since the method measures the depth-averaged horizontal current from ocean surface to the seafloor or the unburied/suspended cable. However, it is expected to measure the differences between depth-averaged currents at different depth using a dedicated vertical multi-layer cable (e.g., Figure S5c) and further estimate the vertical variations of ocean currents. Accordingly, in the revised manuscript we have added corresponding discussions (**Lines 446-465**), as follows:

“In addition, this method measures the depth-averaged horizontal current from ocean surface to the seafloor or the unburied cable. Therefore, measurement of non-uniform 3D currents by this method would be prone to bias (e.g., the biased measurement of linear shear currents exemplified in Figure S7). Yet, this method may have potential for observations of ocean motions with significant vertical components (e.g., submesoscale eddies, internal waves) by utilizing specially planned OF cables. For example, a scheme of multi-layer cable layout as shown in Figure S6c allows for estimation of vertical variations of ocean currents by synchronously measuring the depth-averaged horizontal currents at different layers. It is anticipated that the DAS-based techniques would further facilitate observations of oceanographic phenomena, including tide and internal waves³⁰, and upwellings/downwellings essential for the marine ranch.”

Figure S6c. Schemes of cable layouts designed to retrieve the vertical variations of horizontal ocean currents.

2. “Details on the manual picking and fitting process of the doppler-shifted linear dispersion curves implemented and its limitation are lacking. This is quite relevant, as the parameterization of the f-k transformation (from which the dispersion metrics are obtained) plays a major role on its SNR and its time-space resolution trade-off. As the method strongly relies on some assumptions, namely that: background currents are horizontal; locally steady and uniform; and that the segment layouts are flat (i.e. constant depths), the authors are requested to thoroughly discuss potential biasing induced by non-uniform 3D currents and sloping/complex bathymetry. Additionally, as the exact cable layout at the seafloor is often unknown, the recommendation for proper georeferencing/surveying of the segments should be stressed. Also, some comments on potential effects related to the spatial stability/variability of the directional wave spectrum between the segments would make sense.”

DONE. Thanks for the suggestion. In the revised manuscript, we have provided a more detailed description of the manual picking and fitting process of the doppler-shifted dispersion curves in the Methods section (Lines 542-572), as follows:

“The specific calculation details for this study are described as follows. The cable channels 3875–4075 and 4225–4425 (at 5.5–5.9 km and 6.2–6.6 km respectively; red line segments in Figures 1b and 4a) are selected according to the above selection criterion and their relatively higher signal-to-noise ratios (Figures 2a and d). Both selected cable segments consist of 200 channels and possess respective azimuths $\Phi_1 = 249.7^\circ$ and $\Phi_2 = 214.3^\circ$ (Figure 4a). The DAS observations along each cable segment are transformed into a series of consecutive $f-k$ spectra with both the sliding-window length and step equal to 10 min. The $f-k$ spectra for both cable segments at each time window are then paired for the fitting process. We have developed a dedicated graphical user interface (GUI; Figure S8) based on the Python packages Numpy and Matplotlib for manual fitting process. In the case of given $\Phi_{1,2}$, $h_{1,2}$, $f_{1,2}$ and $k_{1,2}$, this GUI allows for the manual adjustment of the "Current speed" and "Current direction" Sliders to conduct synchronous fitting between the corresponding Doppler-shifted OSGW dispersion curves and the lower edges of the dominant spectral energy packets in the $f-k$ spectra of both cable segments. The water depths $h_{1,2}$ in Eqn. (3) are replaced by the tide-modulated water depths $h'_{1,2}(t)$ (Figure 4d), which are estimated as follows:

$$h'_{1,2}(t) = D_{1,2} + T(t), \quad (4)$$

where $T(t)$ is the tide level variation of the DASO region, with temporal resolution of 10 min after linear interpolation. Due to the lack of exact information of cable layout and burial, the characteristic water depths of the two selected cable segments $D_{1,2}$ are estimated using a specialized GUI (Figure S9). Based on the $f-k$ spectra with inconspicuous Doppler shifts of cable segment x ($x = 1$ or 2) and known $T(t)$, this GUI allows for manual adjustment of the “ D_x ” Slider to control the variation of $h'_x(t)$ to optimize the fit between the corresponding linear OSGW dispersion curves and the lower edges of dominant spectral energy packets. In both GUIs, the dominant spectral energy packets in all of the $f-k$ spectra are highlighted by uniformly rendering with a diverging colormap and filtering out the spectral energy below -115 dB.”

Accordingly, in the supplementary material, we have added two figures showing the graphical user interfaces (GUIs), which are developed for the manual fitting process (Figure S8) and estimation of the characteristic water depths (Figure S9) in this study. The Python code for the GUIs is available on a public data repository (see Code availability).

Figure S8. Graphical user interface (GUI) for manual fitting process of the current-induced Doppler shift. The dashed black curves represent the linear OSGW dispersion curves for channels 3875–4075 and 4225–4425. The solid black curves represent the Doppler-shifted OSGW dispersion curves, changing with the values of current speed U and direction θ , which are controlled by the white handles on the two sliders respectively. The fitting process is conducted by manually sliding the handles to synchronously fit the Doppler-shifted OSGW dispersion curves to the lower edges of dominant spectral energy packets in the $f - k$ spectra of both cable segments. The estimated values of current speed U and direction θ can be saved by clicking the “Save this fit” button. The “Next” button is used for the fitting process of the next spectra pair.

Figure S9. GUI for estimation of the characteristic water depths of cable segments. The $f - k$ spectra of channels 3875–4075 and 4225–4425 with inconspicuous Doppler shifts at four different time periods: 07:30–07:40 UTC on September 13, 15:10–15:20 UTC on September 14, 01:50–02:00 UTC on September 15 and 13:10–13:20 UTC on September 15 are used for estimation. The black curves represent the linear OSGW dispersion curves for the two cable segments, changing with the values of the tide-modulated water depths $h'_{1,2}(t)$. The characteristic water depths D_1 and D_2 are estimated by manually sliding the white handles on the “ D_1 ” and “ D_2 ” Sliders to optimize the fit between the corresponding linear OSGW dispersion curves and the lower edges of dominant spectral energy packets in all the $f - k$ spectra.

Following the reviewer’s suggestion, in the revised manuscript, we further discussed about the impact of varying water depth along the cable (sloping/complex bathymetry) on our proposed method (Lines 336-341), as follows:

“On the other hand, the nonflat cable layout would distort the observed apparent OSGW wavenumbers and therefore the apparent phase velocities. It may also lead to multiple dispersive spectral energy packets with lower edges corresponding with OSGW dispersion curves of diverse water depths, as exemplified in Figures 3a–d. And OSGWs shoaling-up over a slope bathymetry would become higher and steeper, resulting in nonlinear amplitude dispersion effects on the wave phase speeds^{36,37}.”

Furthermore, we envisioned a horizontal cable layout to mitigate such impacts (Lines 431-433), as follows:

“For example, horizontal arrangement of the cable segments as shown in Figure S6a can effectively mitigate the impact of varying bathymetries discussed in Section 3.2.”

Figure S6a. Horizontal arrangement of cable segments designed for mitigating the impact of inconstant water depth.

In addition, following the reviewer’s comment, we discussed the potential biases induced by non-uniform 3D currents by taking linear shear currents (see Figure S7) as an example (Lines 446-450), as follows:

“In addition, this method measures the depth-averaged horizontal current from ocean surface to the seafloor or the unburied cable. Therefore, measurement of non-uniform 3D currents by this method would be prone to bias (e.g., the biased measurement of linear shear currents exemplified in Figure S7).”

Figure S7. Comparison of OSGW dispersion curves with Doppler shifts caused by linear shear currents under different water depths h and surface current speeds U' . The Doppler-shifted OSGW dispersion curves is expressed as $\left(f + \left(1 - \frac{s' \tanh(2\pi kh)}{4\pi k}\right) U' k\right)^2 = \frac{gk}{2\pi} \cdot \tanh(2\pi kh)$, where $s' \in [0, h^{-1}]$ is the unknown constant gradient and $\left(1 - \frac{s' \tanh(2\pi kh)}{4\pi k}\right) U' k$ is the Doppler shift term (Williams *et al.*, 2022). The distribution of the possible Doppler-shifted OSGW dispersion curves depends on the weighting factor of the Doppler shift term $\left(1 - \frac{s' \tanh(2\pi kh)}{4\pi k}\right)$ and is shown as rainbow-colored band. The Doppler shift terms induced by linear shear currents flowing across the two adjacent cable segments with distinct water depths could be significantly different, especially in oceanic settings characterized by shallow bathymetry and rapid surface currents. This would affect the fitting progress and result in biased measurement of current speeds and directions using our method. For comparison, the corresponding linear OSGW dispersion curves are superimposed as gray curves.

We agree with the reviewer on his or her comments “as the exact cable layout at the seafloor is often unknown, the recommendation for proper georeferencing/surveying of the segments should be stressed”. However, due to the security consideration, we were not authorized to survey the layout of the currently operational submarine cable by the operator (China Mobile Communications Group). In the revised manuscript, a brief statement about this has been added (Lines 150-154), as follows:

“The detailed survey of the exact cable layout was not authorized due to security considerations. As an alternative, the water depth along the cable was derived from a single-channel seismic profile that was acquired using a 2000-J Squid 2000 Sparker source (Applied Acoustic Engineering; United Kingdom) onboard a boat traveling at about 5.0 knots.”

In addition, we agree with the reviewer that the spatial stability/variability of the directional wave spectrum between the cable segments would affect the current measurement in this study. In the revised manuscript, we discussed the corresponding impacts (Lines 313-336), as follows:

“Considering directional wave spectra with little energy propagating axially or paraxially down the cable, all OSGWs propagate obliquely across the cable and produce apparent phase velocities faster than those predicted by the linear OSGW dispersion relation.”

The cable selection criterion of this method (Lines 539-541) is anticipated to alleviate such impact.

“The cable segment selection criterion is that the two adjacent segments should orient differently but within the prevailing wave direction range under stable sea states.”

3. “A key point stressed by the authors is that the analysed signals correspond to microseisms. As the frequency band of the most energetic signals considered is between 0.1-0.3 Hz, which is well-known to lie within the secondary microseismic (double-frequency) band, does this imply that all the phenomena observed are caused by non-linear wave-wave interactions? On the other hand, this same frequency band corresponds to the dominant surface gravity wave peak in shallow areas and, as the authors mention, the estimated phase speeds are between 8~12 m/s, which are way too slow for microseisms while are at the same time consistent with typical phase speeds of ocean surface gravity waves. It therefore seems more plausible that all the signals analysed by the authors are being induced directly upon the cable by gravity waves.”

DONE. Yes. We acknowledge that the 0.1–0.3 Hz frequency band is well-known to lie within the secondary (double-frequency) microseismic band. However, these secondary microseisms are often observed on far-field terrestrial seismic networks. In this study, the microseismic noise was observed just beneath the water with water depth shallower than ~30 m when a typhoon passed overhead. We infer a conclusion that the near-field observed microseismic noise in the frequency band of ~0.1–0.3 Hz is directly induced by the ocean surface gravity waves (OSGWs) during Typhoon Muifa (2022), rather than by non-linear wave-wave interactions, mainly from the following issues:

- 1) The dominant frequency band of typhoon-generated ocean waves in shallow waters often spans over ~0.1–0.3 Hz. Consequently, the frequency band of the corresponding induced primary microseisms is also mainly between 0.1 and 0.3 Hz.
- 2) The estimated phase speeds are between 8~12 m/s, consistent with typical phase speeds of OSGWs. Therefore, the microseismic noise could be assumed to originate from the pressure applied by the OSGWs at the seafloor, confirming our judgement of primary microseisms.
- 3) The high-frequency (>0.3 Hz) component is observed tidally modulated, i.e., sensitive to the water depth variations. This is contradictory to the secondary microseisms’ property that they are generated by depth-independent pressure fluctuations on the seafloor induced by non-linear interaction between ocean waves.

In the revised manuscript, we have extended the analysis on the observed microseismic noise, and added a paragraph at the end of subsection 3.1 (Lines 231-247) as follows:

“It is noticeable that although the high-frequency (>0.3 Hz) component seems to lie within the frequency band (~0.1–0.5 Hz) of secondary microseismic noise (SMN), the observed microseismic noise is directly generated by the OSGW seafloor pressure, namely primary microseismic noise (PMN). Traditional observations of microseismic noise on terrestrial seismic networks or OBSs constitute diffuse seismic energy radiated into the far field, whereas the observation is conducted just beneath the water when a typhoon passed overhead in this study. Because the dominant frequency band of typhoon-generated ocean waves in shallow waters can often span over ~0.1–0.35 Hz^{33,34}, the corresponding

near-field PMN can extend beyond 0.3 Hz. In addition, the microseismic noise is observed tidally modulated and sensitive to water depth. This is contradictory to the generation mechanism of SMN⁷, whereby the nonlinear interaction between opposing ocean waves induces a depth-independent pressure fluctuations on the seafloor.”

4. “L 191-6: Although the slowest apparent component is more likely to correspond to the true wavefield speed, this is not always necessarily the case. As an example, what if several swells with different speeds are superposed in a way that those that arrive (nearly) axially are even faster than those at an angle? This is particularly relevant for cyclones, as their directional wave spectrum can be complex and asymmetric among its four quadrants (see e.g. Hu & Chen, 2011-Directional spectra of hurricane-generated waves in the Gulf of Mexico). The fact that previous studies have fitted linear gravity wave dispersion curves to their f-k spectra does not imply that the observed waves propagated parallel to the cable, as f-k dispersion branches could still be apparent. Assuming true propagation along a line is generally incorrect and may generally lead to errors, as only non-axial components might occur. However, the fact that all observed wavelengths are apparent does not appear to affect the results, but the fact should be clarified, specially for the non-specialist readers.”

DONE. Thanks for the reviewer’s constructive suggestions. We acknowledge that it is important to emphasize the property of DAS-observed apparent phase speed. In the revised manuscript, we have modified the text (Lines 304-309) as follows:

“Previous studies^{25,26,28} have indicated that the most prominent feature of the DAS $f-k$ spectrum is that the lower edges of the spectral energy packets mainly fit the linear OSGW dispersion curve³⁵:

$$f^2 = \frac{gk}{2\pi} \cdot \tanh(2\pi kh). \quad (1)$$

Given that the DAS-observed OSGW propagations are apparent, these lower edges are typically thought to be caused by OSGWs propagating axially or paraxially along the cable.”

5. “L 255-7 The authors mention that the picking and fitting is done manually. Although the supplementary video showing the several picked f-k spectra are helpful to visualize this, further details on the workflow and criteria used to perform this picking and fitting are necessary. Also, do the authors envision a potential way to effectively and accurately automatize the procedure?”

DONE. Thanks for these excellent suggestions. In the revised manuscript, we have provided a more detailed description of the manual picking and fitting process of the doppler-shifted dispersion curves in the Methods section (Lines 542-572), as follows:

“The specific calculation details for this study are described as follows. The cable channels 3875–4075 and 4225–4425 (at 5.5–5.9 km and 6.2–6.6 km respectively; red line segments in Figures 1b and 4a) are selected according to the above selection criterion and their relatively higher signal-to-noise ratios (Figures 2a and d). Both selected cable segments consist of 200 channels and possess respective azimuths $\Phi_1 = 249.7^\circ$ and $\Phi_2 = 214.3^\circ$ (Figure 4a). The DAS observations along each cable segment are transformed into a series of consecutive $f-k$ spectra with both the sliding-window length and step equal to 10 min. The $f-k$ spectra for both cable segments at each time window are then paired for the fitting process. We have developed a dedicated graphical user interface (GUI; Figure S8) based on the Python packages Numpy and Matplotlib for manual fitting process. In the case of given $\Phi_{1,2}$, $h_{1,2}$, $f_{1,2}$ and $k_{1,2}$, this GUI allows for the manual adjustment of the "Current speed" and "Current direction" Sliders to conduct synchronous fitting between the corresponding Doppler-shifted OSGW dispersion curves and the lower edges of the dominant spectral energy packets in the $f-k$ spectra of both cable segments. The water depths $h_{1,2}$ in Eqn. (3) are replaced by the tide-modulated water depths $h'_{1,2}(t)$ (Figure 4d), which are estimated as follows:

$$h'_{1,2}(t) = D_{1,2} + T(t), \quad (4)$$

where $T(t)$ is the tide level variation of the DASO region, with temporal resolution of 10 min after linear interpolation. Due to the lack of exact information of cable layout and burial, the characteristic water depths of the two selected cable segments $D_{1,2}$ are estimated using a specialized GUI (Figure S9). Based on the $f-k$ spectra with inconspicuous Doppler shifts of cable segment x ($x = 1$ or 2) and known $T(t)$, this GUI allows for manual adjustment of the “ D_x ” Slider to control the variation of $h'_x(t)$ to optimize the fit between the corresponding linear OSGW dispersion curves and the lower edges of dominant spectral energy packets. In both GUIs, the dominant spectral energy packets in all of the $f-k$ spectra are highlighted by uniformly rendering with a diverging colormap and filtering out the spectral energy below -115 dB.”

Accordingly, in the supplementary material, we have added two figures showing the graphical user interfaces (GUIs), which are developed for the manual fitting process (Figure S8) and estimation of the characteristic water depths (Figure S9) in this study. The Python code for the GUIs is available on a public data repository (see Code availability).

In addition, we further discussed the potential schemes to automatize the procedure (Lines 414-417) as follows:

“Therefore, an algorithm-based automated fitting process, such as one based on digital image processing or machine learning for edge detection and extraction, is crucial to improve the efficiency and accuracy in follow-up work.”

6. “L 258-60 The selection criteria of the three channel ranges shown in Fig 2 was not entirely clear to me until I read these lines. The fact that these kind of specific cable segments are required for the method to work should be mentioned somewhere in the introduction, at least. Furthermore, it is somewhat confusing that none of the data segments highlighted in Figs. 1 and 4 is studied in Fig. 2. It would make sense to at least highlight the segments in Fig. 2.”

DONE. These are excellent suggestions. Following the reviewer’s suggestion, the manuscript was revised to include a brief introduction of the method (Lines 123-143), as follows:

“These observations show that the ocean-bottom DAS-instrumented cable is sensitive to typhoon-induced microseismic noise, even at ultra-low frequencies (<0.1 Hz), and effective in deriving information on OSGW along the cable. We further presented a novel measurement method, which is based on two specific cable segments with different orientations, to successfully reveal both speeds and directions of the horizontal ocean-current component at high spatiotemporal resolutions during such an extreme weather event.”

Also following the reviewer’s comment, we have replotted Figure 2 as follows to add the mean time–frequency spectrogram of the DASO data along channels 4068–4071 (Figure 2d), which are within one of the selected cable segments used for horizontal ocean current component measurement (as shown in Figure 1b and Figure 4a).

Figure 2. Spatial and temporal characteristics of the DASO-recorded microseismic noise during the passage of Typhoon Muifa. (a) Distance–frequency spectrogram of the upper-quartile spectral power at each channel along the cable, with the corresponding water depth (magenta line) overlain. (b) Mean time–frequency spectrogram of the DASO data along channels 1253–1256 (marked by the inverted coral triangle in (a)) for the period from 04:00 UTC on September 13 to 02:00 UTC on September 16, 2022. The overlaid solid coral line represents the tide-modulated water depth $h_{\text{tide}}(t)$ over channels 1253–1256. The dashed black line represents the estimated maximum frequency f_{max} of the OSGW seafloor pressure at channels 1253–1256. (c–e) Same as (b), but for channels 2316–2319, 4068–4071 and 5649–5652 (marked by the green, blue and black inverted triangles in (a) respectively). The solid cyan line in (c) denotes the wind speed observed by the nearby ocean buoy. The solid coral line and dashed black line in (d) represent the same as those in (b) but for channels 4068–4071. The curved gray line in (e) indicates the distance from the centroid of the DASO to the typhoon center, with sizes of the solid circles representing the wind intensity. The two dashed white lines in (c) and (e) indicate the time period when Typhoon Muifa passed over the DASO.

7. “It is known that the detection of surface gravity waves with seafloor DAS is not possible at all depths, as these can attenuate significantly over deep waters, as implied by the eq. referenced in L 138. This poses a major, additional limitation to the method, as it could only be implemented over sections that are not too deep. As far as I see, this limitation is not mentioned and should be clearly stated. For instance, how does the reliability of the method vanishes as a function of depth/wavelength/waveheight? and what are the useful cable depth ranges for the method to work?”

DONE. Thanks for these excellent suggestions. We acknowledge that this method relies on detection of the seafloor pressure perturbation induced by OSGWs with the DAS-instrumented OF cable and can only be implemented over cable sections that are not too deep.

According to the reviewer’s comment, we have added Figure S2 in the supplementary material as follows to better visualize the decrease of seafloor pressure perturbation ΔP induced by an OSGW with water depth.

Figure S2. Decrease of seafloor pressure perturbation ΔP induced by an OSGW of height $\zeta = 1\text{ m}$ with water depth h . The colormap denotes the water depth range. The white curves represent the contours of ΔP when $h = 10, 20, 50, 100, 200$ and 500 m , respectively.

Accordingly, the text was revised as follows:

Lines 198-201

Modified

“This is consistent with the linear theory of OSGWs, whereby the seafloor pressure perturbation ΔP induced by an OSGW of height ζ decreases exponentially with water depth h ; i.e., $\Delta P = \frac{\rho g \zeta}{\cosh(2\pi k h)}$, where ρ is the water density, and g is gravitational acceleration²³.”

to

“This is consistent with the linear theory of OSGWs, whereby the seafloor pressure perturbation ΔP induced by an OSGW of height ζ decreases

exponentially with water depth h (Figure S2), i.e., $\Delta P(h, \zeta) = \frac{\rho g \zeta}{\cosh(2\pi k h)}$, where ρ is the water density and g is gravitational acceleration²⁵.”

In addition, the observed horizontal strain rates E_{DAS} from the pressure perturbation of the OSGWs could be theoretically expressed as $E_{DAS} = \eta_x \cdot \Delta P(h, \zeta)$, where η_x is the normalized horizontal compliance. The η_x could be affected by various factors, including cable burial depth, cable-seafloor coupling, V_p/V_s ratio of sediment and cable construction, resulting in significant variations when subjected to different experimental conditions. Therefore, we are currently unable to provide a specific range of cable depth to which this method can be applied, due to lack of precise data about these factors.

Alternatively, in the revised manuscript we have discussed the depth limitation of this method (Lines 436-444), as follows:

“Given the fact that OSGW-induced seafloor pressure perturbations decay exponentially with water depth, the practical implementation depth H of this method is theoretically capped at one-half of the OSGW wavelength, i.e., $H \leq \pi k^{-1}$. The H is also related to the normalized horizontal compliance η_x of the DAS system, which is the conversion coefficient between the ΔP and horizontal strain rate and affected by various factors²⁸ (e.g., cable-seafloor coupling and cable construction). The H would be deeper for unburied high-sensitivity OF cables than those for telecommunications and power cables used in related studies (e.g., $H \approx 100$ and 150 m reported by *Sladen et al.*²⁵ and *Williams et al.*²⁸, respectively).”

8. “L 305 – Some parts of the methods are not explained in much detail, such that several questions are left open. For instance, what is the selection criteria of the segments? Also, how exactly is the fitting process performed to solve for the system of equations (2)? If, as stated in L324-5, energy below 17dB is filtered for every window, does this imply that the minimum speed dispersion curve envelopes may not be picked or are not being sought at all? If picking is done manually, doesn’t the colormap scale affect the interpreter’s choice? Additionally, details about the uncertainty of the method are missing. This may include a discussion on error propagation during the fitting process, which is prone to ambiguity for noisy/smeared fk images.”

DONE. Following the reviewer’s suggestion, the text has been modified accordingly. The selection criterion for the cable segments have been added in the Methods section as follows:

Lines 539-541

“The cable segment selection criterion is that the two adjacent segments should orient differently but within the prevailing wave direction range under stable sea

states.”

Lines 542-547

“The cable channels 3875–4075 and 4225–4425 (at 5.5–5.9 km and 6.2–6.6 km respectively; red line segments in Figures 1b and 4a) are selected according to the above selection criterion and their relatively higher signal-to-noise ratios (Figures 2a and d).”

The fitting process in this study has been manually conducted using a dedicated graphical user interface (Figure S8), and in the revised manuscript we have added a brief description in Methods section as follows:

Lines 552-557

“We have developed a dedicated graphical user interface (GUI; Figure S8) based on the Python packages Numpy and Matplotlib for manual fitting process. In the case of given $\Phi_{1,2}$, $h_{1,2}$, $f_{1,2}$ and $k_{1,2}$, this GUI allows for the manual adjustment of the "Current speed" and "Current direction" Sliders to conduct synchronous fitting between the corresponding Doppler-shifted OSGW dispersion curves and the lower edges of the dominant spectral energy packets in the $f-k$ spectra of both cable segments.”

The purpose of filtering the spectral energy below 17 dB (with unit of rad/s) for every $f-k$ window is to highlight the dominant spectral energy from OSGWs and neglect the weak energy that is difficult to distinguish from the background instrumental noise. We agree with the reviewer’s comment that “the colormap scale affect the interpreter’s choice”, because the colormap scale may hide the weaker lower edge of the spectral energy packet corresponding to the OSGWs propagating axially along the cable and affect the clarity of the lower edges. After having tried many colormap scale settings, we found that the clarity of the boundary line seems not to be affected by the differences in the colormap scale within a certain range when the signal energy is significantly stronger than the background instrumental noise (as shown in the following figures):

$f-k$ spectra with lower threshold of -115 dB (with a unit of ϵ/s)

$f - k$ spectra with lower threshold of -100 dB (with a unit of ϵ/s)

Nonetheless, selection of cable segments with high signal-to-noise ratios (SNRs) is necessary and the colormap scale should be adjusted to highlight the dominant signal energy. And the selection of colormap also matters. In the revised manuscript, we have added a brief corresponding statement as follows:

Lines 542-547

“The cable channels 3875–4075 and 4225–4425 (at 5.5–5.9 km and 6.2–6.6 km respectively; red line segments in Figures 1b and 4a) are selected according to the above selection criterion and their relatively higher signal-to-noise ratios (Figures 2a and d).”

Lines 570-572

“In both GUIs, the dominant spectral energy packets in all of the $f - k$ spectra are highlighted by uniformly rendering with a diverging colormap and filtering out the spectral energy below -115 dB.”

Following the reviewer’s comment, we further discussed the potential bias associated with the manual fitting process and envisioned a possible automatic way as follows:

Lines 410-417

“Firstly, despite the endeavors undertaken to sharpen the lower edges of dominant energy packets in $f - k$ spectra for more accurate fitting (see details in Methods), it should be acknowledged that human visual perception in the manual fitting process is intrinsically prone to bias, particularly when dealing with dispersion curves with slight Doppler shifts in a smeared $f - k$ spectrum image. Therefore, an algorithm-based automated fitting process, such as one based on digital image processing or machine learning for edge detection and extraction, is crucial to improve the efficiency and accuracy in follow-up work.”

9. “L 313 – The length of each of the selected cable segments is about 400 m (according to the channel spacing reported of 2m), but a constant depth is assigned to each. Is the bathymetry truly flat for these segments? If not (as it appears from the bottom profile), how are these constant depth values defined? and how exactly does the assumption of flat bathymetry under a sloping segment affect the results? This could be e.g. quantified/shown in a figure.”

DONE. Thanks for the good questions. In the revised manuscript, we have adopted the reviewer’s suggestion to discuss the potential impact of inconstant water depths (Lines 336-341), as follows:

“On the other hand, the nonflat cable layout would distort the observed apparent OSGW wavenumbers and therefore the apparent phase velocities. It may also lead to multiple dispersive spectral energy packets with lower edges corresponding with OSGW dispersion curves of diverse water depths, as exemplified in Figures 3a–d. And OSGWs shoaling-up over a slope bathymetry would become higher and steeper, resulting in nonlinear amplitude dispersion effects on the wave phase speeds^{36,37}.”

Furthermore, to alleviate the potential impact of the inconstant water depths along the cable, we have presented a viable solution that involves determining the characteristic water depths of the selected cable segments. Figure S9 showed the specialized GUI for estimations of characteristic water depths. And we envisioned a horizontal cable layout (Figure S6a) to truly avoid such impact as follows:

Lines 431-433

“For example, horizontal arrangement of the cable segments as shown in Figure S6a can effectively mitigate the impact of varying bathymetries discussed in Section 3.2.”

Figure S6a. Horizontal arrangement of cable segments designed for mitigating the impact of inconstant water depth.

10. “A data processing section is missing. Any preprocessing of the data (e.g. unit conversion, filtering, decimation, detrending, etc.) prior to the implementation of the method (or the lack of any processing step) must be reported accordingly.”

DONE. Thanks for these excellent suggestions. Following the reviewer's suggestion, we added a subsection of data processing in Methods (Lines 505-514), as follows:

“Data processing

The raw DAS dataset $DATA_{raw}$ with the unit of phase change rate (rad/s) is first converted to dataset $DATA_{strain\ rate}$ with the unit of strain rate (ϵ/s) as follows:

$$DATA_{strain\ rate} = \frac{\lambda}{4\pi n G \xi} \cdot DATA_{raw}, \quad (2)$$

where $\lambda = 1.550 \times 10^{-9}$ m is the laser wavelength of the DAS system, $n = 1.467$ is refraction index of the OF, $G = 2$ m is the gauge length and $\xi = 0.78$ is the photoelastic scaling factor for longitudinal strain. The $DATA_{strain\ rate}$ is then processed using the Python package Obspy, including: 1) demeaning and detrending; 2) decimating to 2 point per second and 1-Hz low-pass filtering. Noted that the DAS data used in $f - k$ analysis is further decimated to 1 point per second for calculation efficiency.”

And we added a brief description of data processing in the text (Lines 161-162), as follows:

“The raw DAS-recorded phase change data (with a unit of rad/s) was further processed into 2 Hz strain rate data with a unit of ϵ/s (See detailed processing steps in Methods).”

The DAS data used in the former manuscript was raw phase changes and not converted to strain rates. In the revised manuscript, we have converted the phase change rate (rad/s) to strain rate (ϵ/s) and replotted the result Figures 2-4.

Suggested improvements:

11. “As the study deals specifically with the estimation of the magnitude (speed) and azimuth of ocean currents, the proposed title might be a bit too broad and ambiguous. My suggestion would be to change it to “Directional (or azimuthal) monitoring of ocean currents during the passage of Typhoon Muifa (2022) using optical-fiber distributed acoustic sensing” or something similar that emphasizes ocean currents. As there is a recent study on deep sea current detection with DAS using a different approach (hanging cable sections) by Mata Flores et al. (2023), the suggested title would emphasize the fact that your method additionally retrieves current azimuth, as opposed to the former study, that only retrieve magnitudes. Also, including a brief comparison of your method with that of the former study seems appropriate.”

DONE. These are excellent suggestions. We agree with the reviewer's comments “the proposed title might be a bit too broad and ambiguous”. Although both the microseismic noise (subsection 3.1) and ocean wave propagation (subsection 3.2) are

analyzed, the main goal of this manuscript is to investigate the magnitude (speed) and azimuth of ocean currents. Therefore, according to the reviewer's suggestion, the title has been modified from

“Monitoring the sea state during the passage of Typhoon Muifa (2022) using optical-fiber distributed acoustic sensing”

to

“Monitoring ocean currents during the passage of Typhoon Muifa (2022) using optical-fiber distributed acoustic sensing”.

Furthermore, following the reviewer's suggestion, in the revised manuscript we have added a brief comparison of our proposed method with that of *Mata Flores et al. (2023)* and highlighted the retrieval of ocean current directions in our study as follows:

Lines 115-120

“More recently, *Mata Flores et al.*³¹ measured the deep-sea current speed with DAS-recorded vortex-induced vibration of suspended cable segments. However, these DAS-based measurements are solely geared towards ocean current speeds and have not yet been able to estimate the current directions, and the potential of utilizing DAS measurements as dense arrays remains to be further evaluated.”

Lines 125-143

“We further presented a novel measurement method, which is based on two specific cable segments with different orientations, to successfully reveal both speeds and directions of the horizontal ocean-current component at high spatiotemporal resolutions during such an extreme weather event.”

12. “It would encourage the authors to discuss potential hydrodynamic effects on the measurements. For instance, how is the method expected to be affected by the potentially non-linear (high $k\zeta$) wave regime of large storms (such as a typhoons) where linear theory no longer applies and waves can break and interact?”

DONE. Thanks for the excellent suggestions. In the revised manuscript, we have added a discussion about the nonlinear effects caused by potential hydrodynamic activities on current measurements and envisioned a possible solution for this issue as follows:

Lines 417-423

“Secondly, the nonlinearity of OSGWs would bias the ocean current measurements based on this linear-theory-based method. For example, the nonlinear amplitude dispersion effects of OSGWs with high wave steepness ($k\zeta$) over varying shallow bathymetry would lead to higher OSGW phase speeds than those predicted by the linear dispersion relation^{37,47}. So adoption of a proper nonlinear OSGW dispersion relation (e.g., nonlinear Boussinesq theory⁴⁸) and simultaneous deployment of wave height sensors and tide gauges are expected to alleviate the nonlinear effects.”

13. “In Fig 1b, the spatial extent and wind direction of the cyclone would be useful to understand the expected wave directional distribution during the typhoon during closest approach (e.g. contours indicating the extent of the maximum wind speeds/ROCI/isobars/...). Are there any models of wave/wind stress direction available for the region, such as hindcasts?”

DONE. Thanks for the good suggestion. We have searched for the publicly available model results of related atmosphere and ocean parameters. However, no one with enough spatial resolution has been found for the DASO region (~20 km). And we have also attempted to simulate the ocean wave fields of the DASO region using FVCOM and ROMS. Regrettably, due to the lack of precise local topography/bathymetry data, the simulation results were not so satisfactory. Alternatively, we added the local wind observed at the nearby ocean buoy in Figure 1b.

Figure 1. Distributed acoustic sensing observatory (DASO) and track of Typhoon Muifa (2022). (a) Regional map showing the typhoon track (curved black line), with the solid color-coded circles representing the wind intensity. The black star indicates the DASO location. The background image is an FY4A/AGRI satellite image of Typhoon Muifa at 06:15 UTC on September 14, 2022. (b) Local map showing the relative locations of the submarine OF cable (solid blue line with black crosses at a 5-km interval and dashed black line) and typhoon track (curved black line). The black triangle denotes the entry point of the cable into the sea. The purple short lines mark the channel locations corresponding with the spectrograms in Figures 2b-e. The two red segments along the cable are selected for the horizontal current measurements in Section 3.3. The nearby ocean buoy is marked by a green diamond, and the corresponding local wind record during the passage of Muifa is shown as colored feather plot. Typhoon Muifa passed over the DASO at around 14:00 UTC on September 14, 2022. (c) Water depth along the ocean-bottom cable starting from the entry point.

14. “In Fig 2 the authors could use different colors (not repeating them) for each of the curves overlain to improve clarity of the figure. Also, reporting the depths of the shown channel ranges in the figure would help the reader. Including a fourth spectrogram panel for some of the shallow channels around ~5km along the cable (e.g. those used in Fig. 4) might be interesting, as the spectral power seems quite high here.”

DONE. Following the reviewer’s suggestion, we replotted Figure 2 with the curves represented in different colors and an additional spectrogram panel of channels 4068-4071 added as Figure 2d. And to help readers to obtain water depth directly, we further replaced the tidal level curves in Figures 2b and d with corresponding tide-modulated water depth curves. The text has been modified accordingly.

Figure 2. Spatial and temporal characteristics of the DASO-recorded microseismic noise during the passage of Typhoon Muifa. (a) Distance–frequency spectrogram of the upper-quartile spectral power at each channel along the cable, with the corresponding water depth (magenta line) overlain. (b) Mean time–frequency spectrogram of the DASO data along channels 1253–1256

(marked by the inverted coral triangle in (a)) for the period from 04:00 UTC on September 13 to 02:00 UTC on September 16, 2022. The overlaid solid coral line represents the tide-modulated water depth $h_{\text{tide}}(t)$ over channels 1253–1256. The dashed black line represents the estimated maximum frequency f_{max} of the OSGW seafloor pressure at channels 1253–1256. (c–e) Same as (b), but for channels 2316–2319, 4068–4071 and 5649–5652 (marked by the green, blue and black inverted triangles in (a) respectively). The solid cyan line in (c) denotes the wind speed observed by the nearby ocean buoy. The solid coral line and dashed black line in (d) represent the same as those in (b) but for channels 4068–4071. The curved gray line in (e) indicates the distance from the centroid of the DASO to the typhoon center, with sizes of the solid circles representing the wind intensity. The two dashed white lines in (c) and (e) indicate the time period when Typhoon Muifa passed over the DASO.

15. “In general, it would be interesting to see some more discussion on the physical oceanographic implications and applications of the method, at least in a general way. This could optionally focus on the regional oceanographic setting (e.g. well-known surface/deepsea circulations in the region) where the experiment was carried out. For instance, is it possible to link the estimated current speed and azimuth values to previous observations in the region or observations for other cyclones? It is remarkable that tidal modulations and tidal currents can be observed so clearly in the data. As the typhoon-induced currents seem to be canceled or even reversed by the tides, could this mean that some currents might be overshadowed by the local tides if not fast enough? The method could also potentially be used to retrieve tidal components under steady conditions, a fact that is worth highlighting.”

DONE. Thanks for these excellent suggestions. Because this method measures the depth-averaged horizontal currents, it is possible that some relatively weaker currents could be overshadowed by the local tides. Following the reviewer’s suggestion, in the revised manuscript we have added a brief discussion about the potential physical oceanographic applications of this method, as follows:

Lines 444-446

“Therefore, this approach could be potentially used to monitor tidal currents and issue prior warning for extreme tide events (e.g., storm surges) in coastal areas.”

Lines 450-465

“Yet, this method may have potential for observations of ocean motions with significant vertical components (e.g., submesoscale eddies, internal waves) by utilizing specially planned OF cables. For example, a scheme of multi-layer cable layout as shown in Figure S6c allows for estimation of vertical variations of ocean currents by synchronously measuring the depth-averaged horizontal currents at different layers. It is anticipated that the DAS-based techniques would further facilitate observations of oceanographic phenomena, including tide and internal waves³⁰, and upwellings/downwellings essential for the marine ranch.”

Figure S6c. Schemes of cable layouts designed to retrieve the vertical variations of horizontal ocean currents.

16. “This and other DAS methods have a potentially high impact for oceanography, so that cable dedicated deployments might be planned in the future. What would the author’s suggestions be for future studies planning to lay cables targeting ocean currents? For example, in terms of geometry, location, etc? The authors could also highlight that the simultaneous use of multiple cables or additional sensors (e.g. current-meters, buoys, moorings, sensor arrays) might provide a complementary view of the oceanographic regime.”

DONE. Thanks for these excellent suggestions. In the revised manuscript, we have accordingly added some discussions about additional sensors and the dedicated cable layouts that benefit the potential applications of this method in oceanography and envisioned some examples of cable layouts (Figure S6) as follows:

Lines 421-434

“So adoption of a proper nonlinear OSGW dispersion relation (e.g., nonlinear Boussinesq theory⁴⁸) and simultaneous deployment of wave height sensors and tide gauges are expected to alleviate the nonlinear effects. Lastly, it is recommended to install cables that are specifically designed for the purpose of observing ocean currents (e.g., Figures S6a and b), instead of employing pre-existing undersea cables that are rigidly regulated and predetermined in their layout. For example, horizontal arrangement of the cable segments as shown in Figure S6a can effectively mitigate the impact of varying bathymetries discussed in Section 3.2. Schemes of cable layouts shown in Figure S6b are designed to improve the spatial coverage and resolution of ocean current measurements.”

Lines 450-465

“Yet, this method may have potential for observations of ocean motions with significant vertical components (e.g., submesoscale eddies, internal waves) by utilizing specially planned OF cables. For example, a scheme of multi-layer cable layout as shown in Figure S6c allows for estimation of vertical variations of ocean currents by synchronously measuring the depth-averaged horizontal currents at different layers. It is anticipated that the DAS-based techniques would further facilitate observations of oceanographic phenomena, including tide and internal waves³⁰, and upwellings/downwellings essential for the marine ranch.”

Figure S6. Schematic illustration of the cable configuration aimed for measuring ocean currents. (a) Horizontal arrangement of cable segments designed for mitigating the impact of inconstant water depth. (b) Schemes of cable layouts designed to improve the spatial coverage and resolution of ocean current measurements. (c) Schemes of cable layouts designed to retrieve the vertical variations of horizontal ocean currents.

References:

17. “L53-65 – Although some references were included, in my opinion, it would be fair to add a few more on cyclone-related microseisms, considering the vast amount of literature available, including review papers and books.”

DONE. Following the reviewer’s suggestion, in the revised manuscript we have added a few more related references as follows:

“

11. Ebeling, C. W. Chapter One - Inferring Ocean storm characteristics from ambient seismic noise: A historical perspective. in *Advances in Geophysics* (ed. Dmowska, R.) **53**, 1–33 (Elsevier, 2012).
12. Chen, X., Tian, D. & Wen, L. Microseismic sources during Hurricane Sandy. *J. Geophys. Res. Oceans* **120**(9), 6386–6403 (2015).
13. Gualtieri, L. *et al.* The persistent signature of tropical cyclones in ambient seismic noise. *Earth Planet. Sci. Lett.* **484**, 287–294 (2018).
16. Gerstoft, P., Fehler, M. C. & Sabra, K. G. When Katrina hit California. *Geophys. Res. Lett.* **33**(17), L17308 (2006).
20. Retailleau, L. & Gualtieri, L. Multi-phase seismic source imprint of tropical cyclones. *Nat. Commun.* **12**(1), 2064 (2021).

”

18. “L60-5 – The cited reference of Davy *et al.*, 2014 is very relevant for the current work as it dealt with the detection of a cyclone passing overhead an array of broadband OBSs and hydrophones in the ocean, in analogy with the presented case of typhoon Muifa with DAS. However the way that this work is referenced and discussed in the text does not seem make justice to this fact (which is not highlighted at all) nor to its findings in general. For instance, in L64 I find “limited” far more accurate than “prohibited” (see Sec. 6 in Davy *et al.*, 2014). At the same time, I disagree that it was only the sparsity of stations in that work what limited the storm tracking (which in fact I consider quite remarkable given the variable geometry, areal extent and the complexity of the gravity wave and bathymetry-dependent seismic noise radiation patterns of a cyclone, specially in the near-field) and I also disagree that this “...prohibited... a detailed near-field investigations of the TC”, as the authors of that work showed in detail and with good data quality the wave polarization and amplitude attenuation functions induced by the cyclone passing overhead, amongst others.”

DONE. We agree with the reviewer and revised the text as follows:

Lines 82-88

Modified

“Davy *et al.*¹² analyzed the microseismic noise from a network of 57 broadband ocean bottom seismometers (OBSs) to infer the evolution of TC Dumile (2013) over the southwestern Indian Ocean. Although the basic track of TC Dumile was identified, the limited number and density of OBSs (average distance of 200 km

between OBSs) prohibited accurate tracking and detailed near-field investigations of the TC.”

to

“Davy *et al.*⁹ analyzed the microseismic noise from a network of 57 broadband ocean bottom seismometers (OBSs) to infer the evolution of TC Dumile (2013) over the southwestern Indian Ocean, and demonstrated that seafloor microseisms can be used for real-time tracking and monitoring of major storms. However, accurate tracking and detailed near-field investigations of the TC are mainly limited by the complex excitation process and propagation of TC-induced microseisms, as well as the finite number and density of OBSs (average distance of 200 km between OBSs).”

19. “L66-84. A few references could be added that are relevant for surface gravity wave analyses with DAS: Guerin *et al.*, 2022 and Sladen *et al.*, 2019. For ocean microseism studies with DAS: Xiao *et al.*, 2022 and Tonegawa *et al.*, 2022. On the other hand, are there any previous studies addressing surface gravity waves and their relationship with ocean currents from theoretical perspectives or proposed experimental methods not related to DAS, such as acoustic/remote sensing/oceanography? This would be worthy to be mentioned and discussed for general context.”

DONE. Thanks for the suggestion. The following suggested references were added in the revised manuscript.

“

21. Zhan, Z. Distributed acoustic sensing turns fiber-optic cables into sensitive seismic antennas. *Seismol. Res. Lett.* **91**(1), 1–15 (2019).
22. Tonegawa, T. *et al.* Extraction of P wave from ambient seafloor noise observed by distributed acoustic sensing. *Geophys. Res. Lett.* **49**(4), e2022GL098162 (2022).
23. Xiao, H. *et al.* Locating the precise sources of high-frequency microseisms using distributed acoustic sensing. *Geophys. Res. Lett.* **49**(17), e2022GL099292 (2022).
25. Sladen, A. *et al.* Distributed sensing of earthquakes and ocean-solid Earth interactions on seafloor telecom cables. *Nat. Commun.* **10**(1), 5777 (2019).
27. Guerin, G. *et al.* Quantifying microseismic noise generation from coastal reflection of gravity waves recorded by seafloor DAS. *Geophys. J. Int.* **231**(1), 394–407 (2022).

”

Accordingly, the text was modified (Lines 89-92), as follows:

“Distributed acoustic sensing (DAS) is an emerging technology that can provide strain measurements and effectively turn optical-fiber (OF) cables into dense seismo-acoustic arrays²¹, thereby motivating many recent advances in microseism research^{22,23} and ocean observations^{24–30}.”

In addition, according to the reviewer's suggestion, in the revised manuscript we have added discussions about previous studies of the relationship between OSGWs and ocean currents without DAS at the end of subsection 3.2 (Lines 341-369), as follows:

“Additionally, the underlying ocean currents would induce Doppler effects into the OSGW dispersion relationship³⁸⁻⁴¹, which would manifest as non-reciprocal frequency shifts of the spectral energy packets in the DAS $f-k$ spectrum²⁶ (Figures 4b and c). Nevertheless, the Doppler effect can also be used to measure the ocean current.

Ocean current measurements, based on Doppler shifts of OSGW fields, have been successfully conducted using radar and optical systems^{39,40,42-44}. These measurements can characterize the spatial variations of currents, yet they are reliant on the weather. The DAS-based observations of the current-induced Doppler-shifts, on the other hand, have been recently proven effective for measuring current speed²⁸. These new methods, which use the OSGW-generated pressure disturbances at the seafloor, can function effectively even in harsh weather conditions. It is anticipated to offer a more cost-effective means of measuring submesoscale currents than ocean acoustic tomography^{45,46} and conventional oceanographic instrumentation, such as moored current meters and acoustic Doppler current profilers. In the following section, a novel approach utilizing the DAS $f-k$ spectra to characterize ocean currents is proposed. More importantly, even during typhoon conditions, this method can estimate current direction in addition to measuring current speed.”

Accordingly, seven additional relevant reference was added.

“

38. Stewart, R. H. & Joy, J. W. HF radio measurements of surface currents. *Deep-Sea Res.* **21**(12), 1039–1049 (1974).
39. Young, I. R., Rosenthal, W. & Ziemer, F. A three-dimensional analysis of marine radar images for the determination of ocean wave directionality and surface currents. *J. Geophys. Res. Oceans* **90**(C1), 1049–1059 (1985).
40. Dugan, J. P., Piotrowski, C. C. & Williams, J. Z. Water depth and surface current retrievals from airborne optical measurements of surface gravity wave dispersion. *J. Geophys. Res. Oceans* **106**(C8), 16603–17071 (2001).
41. Leckler, F., *et al.* Analysis and interpretation of frequency-wavenumber spectra of young wind waves. *J. Phys. Oceanogr.* **45**(10), 2484–2496 (2015).
42. Parks, A. B. *et al.* HF radar observations of small-scale surface current variability in the Straits of Florida. *J. Geophys. Res.* **114**(C8), C08002 (2009).
43. Lund, B. *et al.* A new technique for the retrieval of near-surface vertical current shear from marine X-band radar images. *J. Geophys. Res. Oceans*, **120**(12), 8466–8486 (2015).
44. Lenain, L. *et al.* Airborne remote sensing of upper-ocean and surface

properties, currents and their gradients from meso to submesoscales. *Geophys. Res. Lett.* **50**(8), e2022GL102468 (2023).

”

20. “L 212-14 – this is a quite well-known fact within the oceanographic community, thus adding a few older references would be very pertinent e.g. Leckler, 2015 - Analysis and Interpretation of Frequency–Wavenumber Spectra of Young Wind Waves”

DONE. Thanks for the suggestion. According to the reviewer’s comment, the text has been revised as follows:

Lines 341-344

Modified

“OSGWs propagating in an ocean current would induce a Doppler effect in the observed signals of the underlying DAS-instrumented cable, which would manifest as non-reciprocal frequency shifts of the spectral energy packets in the DAS $f-k$ spectrum²⁰ (Figures 4b and c).”

to

“Additionally, the underlying ocean currents would induce Doppler effects into the OSGW dispersion relationship³⁸⁻⁴¹, which would manifest as non-reciprocal frequency shifts of the spectral energy packets in the DAS $f-k$ spectrum²⁶ (Figures 4b and c).”

Accordingly, a few older references were added as follows:

“

38. Stewart, R. H. & Joy, J. W. HF radio measurements of surface currents. *Deep-Sea Res.* **21**(12), 1039–1049 (1974).
39. Young, I. R., Rosenthal, W. & Ziemer, F. A three-dimensional analysis of marine radar images for the determination of ocean wave directionality and surface currents. *J. Geophys. Res. Oceans* **90**(C1), 1049–1059 (1985).
40. Dugan, J. P., Piotrowski, C. C. & Williams, J. Z. Water depth and surface current retrievals from airborne optical measurements of surface gravity wave dispersion. *J. Geophys. Res. Oceans* **106**(C8), 16603–17071 (2001).
41. Leckler, F., *et al.* Analysis and interpretation of frequency-wavenumber spectra of young wind waves. *J. Phys. Oceanogr.* **45**(10), 2484–2496 (2015).

”

Clarity and context:

21. “L20 - “...current-induced Doppler shifts”: the authors could indicate what particular physical signal experiences this effect”

DONE. Following the reviewer's comment, the corresponding sentence has been revised as follows:

Lines 26-28

Modified

“The ocean current is also derived via a novel method, based on measurements of the current-induced Doppler shifts.”

to

“Further, a novel method based on the current-induced Doppler shifts of DAS-recorded OSGW dispersions is used to calculate both speeds and directions of horizontal ocean currents.”

22. “L33 - “Tropical cyclones (TCs)”: Perhaps this study could be safely generalized for Cyclones?(this would implicitly also include subtropical and extratropical)”

DONE. Thanks for the suggestions. According to the reviewer's suggestion, the text has been revised as follows:

Lines 42-43

Modified

“Tropical cyclones (TCs) are devastating events that often have tremendous economic and societal impacts.”

to

“Cyclone systems, especially the well-known tropical cyclones (TCs), are devastating events that often have tremendous economic and societal impacts.”

23. “L49 - “Microseismic noise (~0.05–0.5 Hz)...”: Ocean microseismic noise (as e.g. lakes can generate microseisms at up to 2 Hz)”

DONE. Thanks for the suggestions.

Line 55

Modified

“Microseismic noise (~0.05–0.5 Hz)”

to

“Ocean microseismic noise (typically in ~0.05–0.5 Hz)”

24. “L73 – small typo: “optical-fiber””

DONE. Thanks for the correction.

Line 97

Modified

“potical-fiber”

to

“optical-fiber”

25. “L87-9 - “...These observations show that the ocean-bottom DAS-instrumented cable is sensitive to typhoon-induced microseismic noise, even at ultra-low frequencies (<0.1 Hz), and effective in deriving information on OSGW.”. How do the authors tell apart secondary microseisms from SGWs in the frequency range of interest?”

DONE. According to the theory proposed by Longuet-Higgins (1950), secondary microseisms are generated by the pressure fluctuations on the seafloor induced by nonlinear interaction between ocean waves, which have nearly the same frequency and travel in opposite directions. Therefore, secondary microseisms are usually distinguished by the fact that their frequencies are double that of corresponding ocean waves. In this study, because the frequency band of typhoon-generated ocean waves in shallow waters often spans over ~ 0.1 – 0.35 Hz and the DAS-observed microseismic noise is also between 0.1 and 0.3 Hz, we infer a conclusion that the near-field observed microseismic noise is mainly induced by the OSGWs (i.e., primary microseisms), rather than secondary microseisms.

In addition, the estimated phase speeds are between 8 – 12 m/s, consistent with typical phase speeds of OSGWs. Therefore, the observed microseismic noise could be assumed to originate from the pressure applied by the OSGWs at the seafloor, confirming our judgement of primary microseisms. And the high-frequency (>0.3 Hz) component is observed tidally modulated, i.e., sensitive to the water depth variations. This is contradictory to the secondary microseisms’ property that they are generated by depth-independent pressure fluctuations on the seafloor induced by non-linear interaction between ocean waves.

Following the reviewer’s comments, in the revised manuscript, we have extended the analysis on the observed microseismic noise, and added a paragraph at the end of subsection 3.1 (Lines 231-247) as follows:

“It is noticeable that although the high-frequency (>0.3 Hz) component seems to lie within the frequency band (~ 0.1 – 0.5 Hz) of secondary microseismic noise (SMN), the observed microseismic noise is directly generated by the OSGW seafloor pressure, namely primary microseismic noise (PMN). Traditional observations of microseismic noise on terrestrial seismic networks or OBSs constitute diffuse seismic energy radiated into the far field, whereas the observation is conducted just beneath the water when a typhoon passed overhead in this study. Because the dominant frequency band of typhoon-generated ocean waves in shallow waters can often span over ~ 0.1 – 0.35 Hz^{33,34}, the corresponding near-field PMN can extend beyond 0.3 Hz. In addition, the microseismic noise is observed tidally modulated and sensitive to water depth. This is contradictory to the generation mechanism of SMN⁷, whereby the nonlinear interaction between opposing ocean waves induces a depth-independent pressure fluctuations on the seafloor.”

26. “L 108: is the buoy really collocated with the studied cable segments? If not (as it seems), what is the inter-distance for each case?”

DONE. Thanks for the comments. In Supplementary Information, we added a new Figure S1 showing the distance between the buoy and cable channels as follows:

Figure S1. Distance between the ocean buoy and the submarine cable. The four coral dots mark out the distances between the buoy and the four selected channel segments in Figures 2b-e respectively. The two blue line segments denote the distance ranges between the buoy and the selected cable segments in Figure 4.

Accordingly, the text has been revised as follows:

Lines 162-168

Modified

“A collocated ocean buoy provided a continuous and accurate record of the sea-surface winds.”

to

“A nearby ocean buoy (with minimum distance of ~1587 m to the cable; Figure S1) provided a continuous and accurate record of the sea-surface winds.”

27. “L 165: in Fig 2a was the data lowpassed at 0.4Hz? The decrease above this bound seems abnormally sudden”

DONE. We agree with the reviewer’s comment. The Figure 2a in the former manuscript was plotted with 1 Hz DAS data, which was derived by decimating the 500 Hz raw DAS data. The used “decimate” function (provided by Python package Obspy) automatically applied a lowpass filter prior to decimation, which could avoid introducing aliasing artifacts. To avoid the sudden decrease around 0.4 Hz, we have replotted Figure 2 with 2 Hz DAS data as follows:

Figure 2a. Distance–frequency spectrogram of the upper-quartile spectral power at each channel along the cable, with the corresponding water depth (magenta line) overlain.

28. “L 146. The same phenomenon was reported and studied in detail in ref #22.”

DONE. Following the reviewer’s suggestion, the text was revised as follows:

Lines 208-209

Modified

“A similar phenomenon has been observed by *Williams et al.*²¹.”

to

“A similar phenomenon has been observed by *Williams et al.*²⁸ and *Taweasantanon et al.*²⁹.”

29. “L 147-49. Since $kh \sim 1$ seems to be a constrain of the cited equation, is it then only valid for deep waves? Fig 2b is convincing but the authors could also include the same frequency limits for Figs 2c,d”

DONE. Thanks for the reviewer’s comment. Given that the k in this study denotes the wavenumber, and the deep-water limit of the linear wave theory can be expressed as

$\frac{h}{L} = \frac{kh}{2\pi} \geq 0.5$, i.e., $kh \geq \pi$. Thus, the constrain of the cited equation $n = 1/kh \sim 1$ or $kh \sim 1$ is valid for shallow waves in this study.

Following the reviewer’s suggestion, the theoretical maximum frequency of the OSGW seafloor pressure $f_{max}(t)$ limits of the modified Figures 2c and e have been included in Figure S3.

Figure S3. Variations of theoretical maximum frequency of the OSGW seafloor pressure $f_{max}(t)$ under different tide-modulated water depths. The colormap corresponds to water depth at low-tide time $h_{low-tide}$, and the tide-modulated water depth is derived by simply adding $h_{low-tide}$ and tide level up. The four white curves represent the $f_{max}(t)$ when $h_{low-tide} = 8, 10, 15$ and 20 m, respectively.

30. “L 158: The authors could comment somewhere on the specific correlation method used to obtain the coefficient. The same applies to figures S1 and S2.”

DONE. Following the reviewer’s suggestion, we have added the descriptions of correlation methods in the captions of Figures S4 and S5 as follows:

Figure S4. Correlation between DASO spectrogram and buoy-observed wind speed. (a, c) Mean spectrogram at channels 2316–2319 and 5649–5652 respectively. The solid cyan line denotes the wind speed observed by the collocated ocean buoy. (b, d) Correlation coefficients between the mean spectrogram and observed wind speed at different frequencies. The correlation coefficients $R(f)$ between DAS-recorded spectrogram S and buoy-observed surface wind speed W was calculated as $R(f) = \text{corcoef}[S(f), W]$, where f is the frequency and corcoef is a function of Python package Numpy for correlation coefficients calculation. Then the maximum Pearson correlation coefficients R_{max} and corresponding frequency f_R is derived by

$$\begin{cases} R_{max} = \max R(f) \\ f_R = \text{argmax} R(f) \end{cases}$$

Figure S5. Correlation of measured ocean current directions (solid dark blue line) and buoy-observed wind directions (solid coral line). The correlation coefficient R between measured ocean current directions C_{dir} and buoy-observed surface wind directions W_{dir} was calculated as $R = \text{corroef}[C_{dir}, W_{dir}]$, where `corroef` is a function of Python package Numpy for correlation coefficients calculation.

31. “L 183. “with a northeast component” would be more accurate. A similar comment applies to L185”

DONE. Thanks for these suggestions.

Lines 275-279

Modified

“mainly originates from OSGWs propagating to the northeast across the cable toward the coast of Daishan Island (landward); conversely, the spectral energy in the left quadrant, which possesses a negative phase velocity, is generated by OSGWs propagating to the southwest (seaward).”

to

“mainly originates from OSGWs propagating with a northeast component across the cable toward the coast of Daishan Island (landward); conversely, the spectral energy in the left quadrant, which possesses a negative phase velocity, is generated by OSGWs propagating with a southwest component (seaward).”

32. “L 186-190: As the typhoon approaches from the south, westward winds occur just before its arrival, while eastward winds dominate afterwards, which is contrary to what is written in the text. This can be confirmed by the dominant seaward energy in Fig 3b and the opposite in 3c.”

DONE. Thanks for the comment. The “northeast winds and southwest winds” in the text follows the wind direction definition, which denotes the wind blow FROM northeast TO southwest and that blow FROM southwest TO northeast, and has the same meaning as the reviewer’s comment. For easier understanding, we followed the reviewer’s comment to modify the description of wind direction as where it blows towards in the text.

Lines 282-294

Modified

“resulting in predominantly northeast wind and southwest winds blowing over the water when Typhoon Muifa is approaching and departing the DASO”

to

“resulting in predominantly southwestward winds and northeastward winds blowing over the water when Typhoon Muifa is approaching and departing the DASO”

33. “L198-200: I fail to understand the exact connection of this sentence to the previous ones: the fact DAS can provide measurements of surface gravity waves has already been demonstrated.”

DONE. Thanks, we have **deleted** the confusing sentence.

34. “L211 – what other factors?”

DONE. Thanks for the comments. In the revised manuscript we have added the other factors.

Lines 310-313

Modified

“The fit between the lower edge of the spectral energy packet and the linear OSGW dispersion curve could be affected by several factors, including the ocean current along the cable.”

to

“However, the fit between the lower edge of the spectral energy packet and the linear OSGW dispersion curve could be affected by several factors, including (1) the directional wave spectrum, (2) the varying bathymetry along the cable, and (3) the ocean current fields above the cable.”

35. “L 252-3 – what are the specific requirements for the method to work on the alluded environments?”

DONE. Thanks for the comments. In the revised manuscript, we have added the selection criterion for the cable segments as a specific requirement for the method as follows:

Lines 539-541

“The cable segment selection criterion is that the two adjacent segments should orient differently but within the prevailing wave direction range under stable sea states.”

36. “In Figure 3, between ~4-6 km along the cable there is a shallow cable section of high wave energy which also increases as the storm approaches, both seawards and landwards.

However, the wave speeds appear to increase a bit more in the seaward component relative to the background level than they do after the storm leaves in the landward component. What could cause this?”

DONE. We agree with the reviewer on his or her meticulous comments. The apparent phase velocities of seaward OSGWs as Muifa approaches are strengthened a bit more than those of landward OSGWs during the departure of Muifa at the first ~6 km of the cable. This is likely relevant with the incident angles of the OSGWs propagating across the cable. According to the observed local wind directions (Figure 1b), after the passage of Muifa, the landward OSGWs are expected to propagate across the first ~6 km of the cable at relatively smaller incident angles, which yield relatively slower apparent phase velocities.

37. “L 312 The wavelengths retrieved from a DAS segment are always apparent, meaning that the wavenumbers (k_1, k_2) are apparent too. This should be emphasized, as the method would eventually not allow for the recovery of the propagation angle of the waves. Some additional tests or discussion on the expected performance of the method within the parameter domain would help in its validation, i.e. as a function of the amplitude and angle of incidence of the waves relative to the cable, the relative angle between the segments, the water depth, the strength and azimuth of the current, etc.”

DONE. Thanks for these great suggestions. In the revised manuscript, we have accordingly emphasized that the wavenumbers $k_{1,2}$ is apparent as follows:

Lines 537-538

Modified

“and $f_{1,2}$ and $k_{1,2}$ are the frequencies and wavenumbers of the two corresponding $f-k$ spectra, respectively”

to

“and $f_{1,2}$ and $k_{1,2}$ are the frequencies and apparent wavenumbers of the two corresponding $f-k$ spectra, respectively”

Following the reviewer’s suggestion, in the revised manuscript we have discussed how the effectiveness of this method changes with the wave amplitude and water depth as follows:

Lines 198-201

“This is consistent with the linear theory of OSGWs, whereby the seafloor pressure perturbation ΔP induced by an OSGW of height ζ decreases exponentially with water depth h (Figure S2), i.e., $\Delta P(h, \zeta) = \frac{\rho g \zeta}{\cosh(2\pi k h)}$, where ρ is the water density and g is gravitational acceleration²⁵.”

Lines 436-444

“Given the fact that OSGW-induced seafloor pressure perturbations decay exponentially with water depth, the practical implementation depth H of this

method is theoretically capped at one-half of the OSGW wavelength, i.e., $H \leq \pi k^{-1}$. The H is also related to the normalized horizontal compliance η_x of the DAS system, which is the conversion coefficient between the ΔP and horizontal strain rate and affected by various factors²⁸ (e.g., cable-seafloor coupling and cable construction). The H would be deeper for unburied high-sensitivity OF cables than those for telecommunications and power cables used in related studies (e.g., $H \approx 100$ and 150 m reported by *Sladen et al.*²⁵ and *Williams et al.*²⁸, respectively).”

38. “L 319 – How are this tide-modulated depths obtained? Is a reference tide gauge required for the method to work? If so, this should be clearly stated.”

DONE. Following the reviewer’s comments, we have modified the text to add the estimation of tide-modulated depths as follows:

Lines 557-570

“The water depths $h_{1,2}$ in Eqn. (3) are replaced by the tide-modulated water depths $h'_{1,2}(t)$ (Figure 4d), which are estimated as follows:

$$h'_{1,2}(t) = D_{1,2} + T(t), \quad (4)$$

where $T(t)$ is the tide level variation of the DASO region, with temporal resolution of 10 min after linear interpolation. Due to the lack of exact information of cable layout and burial, the characteristic water depths of the two selected cable segments $D_{1,2}$ are estimated using a specialized GUI (Figure S9). Based on the $f-k$ spectra with inconspicuous Doppler shifts of cable segment x ($x = 1$ or 2) and known $T(t)$, this GUI allows for manual adjustment of the “ D_x ” Slider to control the variation of $h'_x(t)$ to optimize the fit between the corresponding linear OSGW dispersion curves and the lower edges of dominant spectral energy packets.”

In addition, reference tide gauges are necessary for improving the precision of the method. Thanks for the suggestion. We added the corresponding statement in the revised manuscript as follows:

Lines 421-423

“So adoption of a proper nonlinear OSGW dispersion relation (e.g., nonlinear Boussinesq theory⁴⁸) and simultaneous deployment of wave height sensors and tide gauges are expected to alleviate the nonlinear effects.”

39. “The seismic profile/bathymetry data shown in Fig 1c does not seem to be mentioned in the Data availability statement.”

DONE. Thanks for the comments. In the revised manuscript, we have added the statement of seismic profile data in Data availability as follows:

Lines 585-587

“The whole datasets of DAS, seismic profile and ocean buoy observations are

available from the corresponding author on request.”

40. “The gauge length, time sampling rate and unit measured by the DAS system (strain/strain rate?) are missing and should be reported.”

DONE. Thanks for the suggestion. In the revised manuscript we have added the corresponding descriptions as follows:

Lines 155-162

Modified

“The DAS system, which was developed by the University of Science and Technology of China, was configured to acquire data at a 2-m channel spacing along the cable, creating 9780 simultaneously recording sensors. However, the first 1125 channels are subaerial on the Daishan island, leaving 8655 channels (corresponding to 17.3 km of the cable; solid blue line in Figure 1b) distributed beneath the sea surface.”

to

“The DAS system, which was developed by the University of Science and Technology of China, was configured to probe phase changes of Rayleigh backscattering with 500-Hz sampling rate and 2-m gauge length. The used cable was spatially sampled at a 2-m channel interval along the cable, creating 9780 simultaneously recording sensors. However, the first 1125 channels are subaerial on the Daishan island, leaving 8655 channels (corresponding to 17.3 km of the cable; solid blue line in Figure 1b) distributed beneath the sea surface. The raw DAS-recorded phase change data (with a unit of rad/s) was further processed into 2 Hz strain rate data with a unit of ϵ/s (See detailed processing steps in Methods).”

Reviewer #2

“This is a review of the manuscript "Monitoring the sea state during the passage 1 of Typhoon Muifa (2022) using optical-fiber distributed acoustic sensing". This manuscript presents a very nice observation of the effect on Distributed Acoustic Sensing of a typhoon propagation over a fiber optic cable. This is an exceptional setting to study the seismic signals generated by strong oceanic events. I do not have many comments and I found this study very interesting but I believe a few things should be considered.”

THANK YOU. We appreciate very much for these helpful suggestions. The text has been revised accordingly. And the revision has significantly improved the manuscript.

1. “While I understand the paper is observational, I think the authors should discuss at least a little the waves that are observed on the signals. There is no real discussion of this while I feel necessary since phase velocities are discussed. One disadvantage of DAS data is that observations are only made over one dimension. Consequently, a discussion of wave propagation and types cannot be avoided in my opinion. The authors show very nice observations but the impact could be discussed more.”

DONE. Thanks for the great suggestion. In the revised manuscript, we have extended the analysis on the observed microseismic noise, and added a paragraph at the end of subsection 3.1 (Lines 231-247) as follows:

“It is noticeable that although the high-frequency (>0.3 Hz) component seems to lie within the frequency band (~ 0.1 – 0.5 Hz) of secondary microseismic noise (SMN), the observed microseismic noise is directly generated by the OSGW seafloor pressure, namely primary microseismic noise (PMN). Traditional observations of microseismic noise on terrestrial seismic networks or OBSs constitute diffuse seismic energy radiated into the far field, whereas the observation is conducted just beneath the water when a typhoon passed overhead in this study. Because the dominant frequency band of typhoon-generated ocean waves in shallow waters can often span over ~ 0.1 – 0.35 Hz^{33,34}, the corresponding near-field PMN can extend beyond 0.3 Hz. In addition, the microseismic noise is observed tidally modulated and sensitive to water depth. This is contradictory to the generation mechanism of SMN⁷, whereby the nonlinear interaction between opposing ocean waves induces a depth-independent pressure fluctuations on the seafloor.”

2. “I had to go back and forth a lot to use the figures and I think a few changes could be made to help. In Figure 2 b,c,d the locations are defined as channels but in 2 a and Figure 1 the distance is represented in km. Moreover, the water depth is an important feature so I would make that more directly clear for the different cases.”

DONE. These are excellent suggestions. Following the reviewer’s suggestion, we have replotted Figure 2 as follows. The text has been modified accordingly.

Figure 2. Spatial and temporal characteristics of the DASO-recorded microseismic noise during the passage of Typhoon Muifa. (a) Distance–frequency spectrogram of the upper-quartile spectral power at each channel along the cable, with the corresponding water depth (magenta line) overlain. (b) Mean time–frequency spectrogram of the DASO data along channels 1253–1256 (marked by the inverted coral triangle in (a)) for the period from 04:00 UTC on September 13 to 02:00 UTC on September 16, 2022. The overlaid solid coral line represents the tide-modulated water depth $h_{\text{tide}}(t)$ over channels 1253–1256. The dashed black line represents the estimated maximum frequency f_{max} of the OSGW seafloor pressure at channels 1253–1256. (c–e) Same as (b), but for channels 2316–2319, 4068–4071 and 5649–5652 (marked by the green, blue and black inverted triangles in (a) respectively). The solid cyan line in (c) denotes the wind speed observed by the nearby ocean buoy. The solid coral line and dashed black line in (d) represent the same as those in (b) but for channels 4068–4071. The curved gray line in (e) indicates the distance from the centroid of the DASO to the typhoon center, with sizes of the solid circles representing the wind intensity. The two dashed white lines in (c) and (e) indicate the time period when Typhoon Muifa passed over the DASO.

3. “It would be interesting to represent the location of the particular focus zones showed on Figure 2 also in Figure 1. Moreover, the authors show how the tidal signals dominate in shallow water on one side of the fiber. This was quite surprising and interesting for me. I believe it would be useful to show the observations on the other side of the fiber. Indeed, as the slope is very different, a comparison would be very informative.”

DONE. According to the reviewer’s comments, we marked the locations of the cable channels shown in Figures 2b-e with four purple short lines in Figure 1b as follows:

Figure 2b. Local map showing the relative locations of the submarine OF cable (solid blue line with black crosses at a 5-km interval and dashed black line) and typhoon track (curved black line). The black triangle denotes the entry point of the cable into the sea. The purple short lines mark the channel locations corresponding with the spectrograms in Figures 2b-e. The two red segments along the cable are selected for the horizontal current measurements in Section 3.3. The nearby ocean buoy is marked by a green diamond, and the corresponding local wind record during the passage of Muifa is shown as colored feather plot. Typhoon Muifa passed over the DASO at around 14:00 UTC on September 14, 2022.

In addition, as the cable distance increases, the microseismic noise observed by the DASO is subject to greater contamination from the instrumental noise of the DAS system. Consequently, the signal-to-noise ratio of the DAS observation turns really low on the other end of the fiber. Following the reviewer’s comment, as an alternative, we have added the spectrogram of shallow cable channels 4068-4071 (at ~5.89 km) in Figure 2d. These cable channels are distributed with similar depths/slopes with those on the other end of the fiber. The corresponding spectrogram shows similar spectral patterns with the spectrogram of channels 1253–1256 in Figure 2b.

Clarity and context:

4. “Line 17: I am not sure I would use the term “hydrological” in the first sentence since it suggests that is what DAS data obtains. I am not sure it is there yet.”

DONE. Following the reviewer’s comment, we **deleted** the term “hydrological”.

5. “Line 33-36: The authors mention the damage on land and need for intensity estimation at the beginning of the paper. To me this implied that there was estimation in the manuscript of intensity on land which I realized later was not the case.”

DONE. We agree with the reviewer on the possible misunderstanding. According to the reviewer’s comments, in the revised manuscript we have modified the corresponding sentences in the Introduction section as follows:

Lines 43-50

Modified

“However, accurately determining TC intensity (defined by the maximum sustained sea-surface wind speed) is still a challenge due to the lack of sufficient *in situ* observations¹. While the Dvorak technique^{2,3}, which is empirically based on cloud patterns and the infrared cloud-top temperature, is widely considered the best available tool to determine TC intensity from satellite imagery, its accuracy is unavoidably contaminated by rainfalls, clouds, breaking waves, and spray⁴. Furthermore, the inherently subjective nature of the Dvorak method means that significant discrepancies often arise among different forecasters and agencies, thereby contributing to recent vigorous debate on the link between TC intensification and global warming⁵⁻⁷. *In situ* measurements, such as aircraft reconnaissance and ocean buoys, are effective in reducing TC intensity and position uncertainties⁸.”

to

“However, accurately determining TC development, particularly TC intensity, is still a challenge due to the lack of sufficient *in situ* observations¹. While the Dvorak technique^{2,3}, which is empirically based on cloud patterns and the infrared cloud-top temperature, is widely considered the best available tool to determine TC intensity from satellite imagery, its accuracy is unavoidably contaminated by rainfalls, clouds, breaking waves, and spray⁴. *In situ* measurements, such as aircraft reconnaissance and ocean buoys, are effective in reducing TC intensity and position uncertainties⁵.”

6. “Line 89 : Frequencies inferior to 0.1 Hz are not “ultra low” in seismology. I would suppress the ultra.”

DONE. YES. Following the reviewer’s suggestion, we have modified “ultra-low” to “low” (**Line 124**).

7. “Figure 2: It could help to add the colored triangles from a) in b,c,d to make the reading of the figure easier”

DONE. Thanks for the suggestion. We modified the Figure 2 accordingly.

8. “Line 229-230: I understand that the authors detail the methodology in the method part but something should still be said about it apart from the fact that it is “novel”.”

Done. These are really great suggestions. According to the reviewer’s suggestion, we have added a paragraph at the end of subsection 3.2 (**Lines 346-369**) to compare the DAS-based current measurement method with some related previous studies, as follows:

“Ocean current measurements, based on Doppler shifts of OSGW fields, have been successfully conducted using radar and optical systems^{39,40,42-44}. These measurements can characterize the spatial variations of currents, yet they are reliant on the weather. The DAS-based observations of the current-induced Doppler-shifts, on the other hand, have been recently proven effective for measuring current speed²⁸. These new methods, which use the OSGW-generated pressure disturbances at the seafloor, can function effectively even in harsh weather conditions. It is anticipated to offer a more cost-effective means of measuring submesoscale currents than ocean acoustic tomography^{45,46} and conventional oceanographic instrumentation, such as moored current meters and acoustic Doppler current profilers. In the following section, a novel approach utilizing the DAS $f-k$ spectra to characterize ocean currents is proposed. More importantly, even during typhoon conditions, this method can estimate current direction in addition to measuring current speed.”

REVIEWERS' COMMENTS

Reviewer #1 (Remarks to the Author):

Major comments:

- The authors have properly addressed the concerns raised during the first review. I am glad to find a great effort and significant improvements in this new version. There are only some minor comments that I mention below that the authors could easily address.
- I think that the addition of the proposed cable layouts to extend the detection capacity of the method to 3D currents is interesting and valuable. It would be nice to see the results of such dedicated layouts somewhere in the future, if planned.
- The shared tool for manual picking of the best-fitting dispersion curves is a creative and valuable contribution to the community.
- The additional figures and text contribute to clarify the content of the manuscript. The method seems promising and straight-forward to obtain first-order estimates of horizontal ocean currents in shallow water environments

Minor comments:

- L465: "...for the marine branch"?
- L282-294: It is true that the convention to name the winds according to their direction of origin is indeed broadly used in meteorology. I apologize for the confusion. The authors may decide on whether to stick to the original (and probably more correct) wind direction convention of their first manuscript, or to use the less common one adopted in the revised one.

Inflammatory material: None

Reviewer #2 (Remarks to the Author):

This is a second review for the manuscript "Monitoring ocean currents during the passage of Typhoon Muifa (2022) using optical-fiber distributed acoustic sensing". After reading the answers from the authors to the comments made by myself and the other reviewer, and the new version of the manuscript, I believe the authors have answered them thoroughly. I think the manuscript is now ready for publication.

Point-by-Point Response to Comments by the Editor and Reviewers

Reviewer #1

Major comments:

- “The authors have properly addressed the concerns raised during the first review. I am glad to find a great effort and significant improvements in this new version. There are only some minor comments that I mention below that the authors could easily address.”
- “I think that the addition of the proposed cable layouts to extend the detection capacity of the method to 3D currents is interesting and valuable. It would be nice to see the results of such dedicated layouts somewhere in the future, if planned.”
- “The shared tool for manual picking of the best-fitting dispersion curves is a creative and valuable contribution to the community.”
- “The additional figures and text contribute to clarify the content of the manuscript. The method seems promising and straight-forward to obtain first-order estimates of horizontal ocean currents in shallow water environments.”

THANK YOU. We appreciate very much for the meticulous review. We have incorporated all the comments by Reviewer 1.

Minor comments:

1. “L465: “...for the marine branch”?”

DONE. Thanks for the comment. We apologize for using inaccurate terminology. It should be “marine ranching”.

In the revised manuscript, we have modified

“...for the marine ranch.”

to

“...for the marine ranching.”

2. “L282-294: It is true that the convention to name the winds according to their direction of origin is indeed broadly used in meteorology. I apologize for the confusion. The authors may decide on whether to stick to the original (and probably more correct) wind direction convention of their first manuscript, or to use the less common one adopted in the revised one.”

DONE. Thanks for the suggestion. We decided to stick to the original wind direction convention in our first manuscript.

In the revised manuscript, we have accordingly modified

“resulting in predominantly southwestward winds and northeastward winds blowing over the water when Typhoon Muifa is approaching and departing the DASO”

to

“resulting in predominantly northeast winds and southwest winds blowing over the water when Typhoon Muifa is approaching and departing the DASO”.

Reviewer #2

“This is a second review for the manuscript "Monitoring ocean currents during the passage of Typhoon Muifa (2022) using optical-fiber distributed acoustic sensing". After reading the answers from the authors to the comments made by myself and the other reviewer, and the new version of the manuscript, I believe the authors have answered them thoroughly. I think the manuscript is now ready for publication.”

THANK YOU. We appreciate very much for the reviewer's helpful suggestions during the first review.